# PolyPose: Deformable 2D/3D Registration via Polyrigid Transformations

**Vivek Gopalakrishnan**
MIT
vivek@csail.mit.edu

**Neel Dey**
MIT, MGH, and HMS
ndey@mgh.harvard.edu

**Polina Golland**
MIT
polina@csail.mit.edu

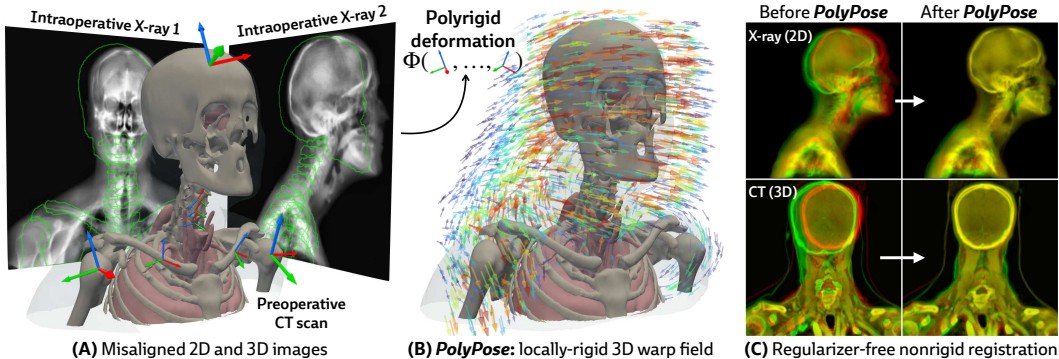

Figure 1: **PolyPose is a locally-rigid framework for sparse-view deformable 2D/3D registration.**
**(A)** PolyPose can deformably align a high-resolution preoperative 3D volume to as few as two intraoperative 2D X-rays without the need of expensive regularizers or hyperparameter optimization. **(B)** To tackle this highly ill-posed problem, we estimate the poses (⌐) of rigid bodies in the volume and smoothly interpolate them in space to produce a topologically consistent locally-rigid warp. **(C)** Using the estimated warps, PolyPose provides 3D volumetric guidance to procedures where only minimal supervision is available from intraoperative 2D X-rays.

## Abstract

Determining the 3D pose of a patient from a limited set of 2D X-ray images is a critical task in interventional settings. While preoperative volumetric imaging (e.g., CT and MRI) provides precise 3D localization and visualization of anatomical targets, these modalities cannot be acquired during procedures, where fast 2D imaging (X-ray) is used instead. To integrate volumetric guidance into intraoperative procedures, we present PolyPose, a simple and robust method for deformable 2D/3D registration. PolyPose parameterizes complex 3D deformation fields as a composition of rigid transforms, leveraging the biological constraint that individual bones do not bend in typical motion. Unlike existing methods that either assume no inter-joint movement or fail outright in this under-determined setting, our polyrigid formulation enforces anatomically plausible priors that respect the piecewise-rigid nature of human movement. This approach eliminates the need for expensive deformation regularizers that require patient- and procedure-specific hyperparameter optimization. Across extensive experiments on diverse datasets from orthopedic surgery and radiotherapy, we show that this strong inductive bias enables PolyPose to successfully align the patient's preoperative volume to as few as two X-rays, thereby providing crucial 3D guidance in challenging sparse-view and limited-angle settings where current registration methods fail. Additional visualizations, tutorials, and code are available at https://polypose.csail.mit.edu.

39th Conference on Neural Information Processing Systems (NeurIPS 2025).

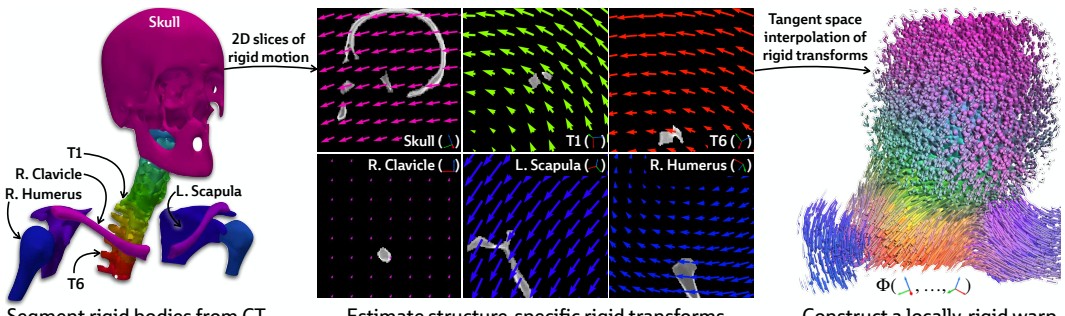

Figure 2: **Illustration of polyrigid deformation fields.** We visualize 2D slices of the rigid motion acting on every articulated structure. Linearly combining these transforms in the tangent space yields a smooth and invertible deformation field, which we color by the relative contribution from every structure. PolyPose enables the recovery of this 3D deformation field via differentiable rendering.

# 1 Introduction

Estimating the 3D position of anatomical structures from 2D X-ray images is a critical task for clinical interventions that require millimeter-level precision, such as image-guided surgery [1–5] or the delivery of radiotherapy in cancer treatment [6–10]. The number of 2D X-rays available for 3D volumetric pose estimation is directly proportional to the radiation exposure to the patient and clinical team, as well as the time available for the procedure, thereby reducing the number of X-rays acquired [11, 12]. Furthermore, the geometry of X-ray scanners limits the angular range of acquisitions, introducing spatial ambiguities along the projection direction and challenges for 3D localization [13]. While patients undergoing surgery and radiotherapy typically have previously acquired 3D volumes, such as computed tomography (CT) scans, their use is confounded by their misalignment with the intraoperative 2D X-rays as patients move between acquisitions (see the misaligned outlines in Figure 1A).

Several parameterizations of 2D/3D motion have been proposed to align these modalities. For example, rigid 2D/3D registration methods align global structure [14–17], but do not account for the soft tissue deformation or articulated inter-joint motion that occurs during procedures and creates localization challenges. Other work estimates point-wise displacement fields using either deep learning [18–22] or optimization [23–25]. However, given the minimal supervision available for estimating 3D deformations in 2D sparse-view and limited-angle settings, deformable models require extensive application-specific regularization to generate anatomically faithful warps, thereby introducing new modeling decisions and hyperparameter tuning for every subject, procedure, and anatomical region.

Our approach is instead motivated by a generic anatomical prior: bones are rigid bodies. We parameterize deformable 2D/3D registration using a low-dimensional *polyrigid* model with limited degrees of freedom (Figure 2), where transformations are composed from individually estimated rototranslations of multiple articulated structures that are linearly combined in the tangent space $\mathfrak{se}(3)$ [26]. This reduces the number of optimizable parameters from the order of voxels in the CT volume to the order of the number of rigid components. Furthermore, unlike other low-dimensional deformation models (e.g., splines [27] or linear bases [18]), polyrigid transforms have several desirable properties by construction, such as smoothness, invertibility, and coordinate frame invariance [26].

Our method, PolyPose, enables the estimation of highly accurate non-rigid deformations that are anatomically plausible and topologically consistent. We do this via differentiable X-ray rendering, providing piece-wise 2D/3D registration targets from which to construct a polyrigid warp. Empirically, across diverse datasets, PolyPose is robust even for a small number of input views from limited angles. Furthermore, given its strong inductive priors, PolyPose does not require any regularization and has no tunable hyperparameters other than the step size of the optimizer. Our method outperforms both deep learning and optimization-based 2D/3D registration methods and enables the 3D localization of critical structures during medical interventions from intraoperative 2D images.

**Contributions.** To summarize, PolyPose contributes:

- A regularization-free framework for deformable 2D/3D registration that estimates polyrigid deformation fields using differentiable X-ray rendering.

- A hyperparameter-free weighting function for linearly combining multiple rigid transformations, providing out-of-the-box generalization to new surgical and therapeutic procedures.
- An anatomically motivated motion model that is robust in sparse-view and limited-angle settings and produces smooth, invertible, and accurate deformation fields by construction.

## 2   Related Work

**Rigid 2D/3D registration.** Given a 2D X-ray and a 3D CT volume, rigid registration methods estimate a global rigid transformation in $\mathbf{SE}(3)$ that optimally aligns the two images [28, 29]. While state-of-the-art methods can now determine the pose of rigid bodies with less than a millimeter of error [15, 16] (which, in a different reference frame, is equivalent to estimating the extrinsic matrix of the image), they fail to describe the motion of volumes subject to non-rigid deformable transformations.

**Deformable 2D/3D registration.** Non-rigid deformable 2D/3D registration is crucial to radiation oncology, where a dense displacement field is needed to align a preoperative planning CT volume with multiple intraoperative X-ray images [20, 23]. As deformably aligning a 3D volume to a set of sparse 2D X-rays is severely ill-posed, deformable 2D/3D methods rely on complex regularization schemes (e.g., diffusion [30], total variation [31], elastic penalties [32]), introducing numerous hyperparameters that must be carefully tuned for every procedure, subject, and anatomical region.

**Deformable 3D/3D registration.** Many methods exist to reconstruct 3D cone-beam computed tomography (CBCT) volumes from multiple 2D X-rays [33]. As such, one could reformulate multi-view 2D/3D registration as a 3D/3D registration task, an active research area, and use recent foundation models for multimodal 3D/3D registration [34–36] or improved solvers for iterative deformable 3D/3D registration [37–39]. Unfortunately, the reconstructed CBCTs produced from sparse ($< 10$) X-rays have very low SNR and suffer severe streaking artifacts [40, 41], complicating their use as registration targets. In parallel, the broader vision literature has proposed several alternative representations of 3D deformation fields for large deformations. For instance, methods such as Nerfies [32] and RAFT-3D [42] estimate dense $\mathbf{SE}(3)$ fields in which each spatial location is assigned an independent rigid transformation. While expressive, these dense deformation models are severely underconstrained in clinical settings characterized by sparse-view and limited-angle X-ray acquisitions.

**Learning-based deformable 2D/3D registration.** To avoid solving an expensive optimization problem for every new pair of 2D X-rays and 3D volume, numerous deep learning methods have been proposed for deformable 2D/3D registration. For example, methods like LiftReg [18] and 2D3D-RegNet [19] rely on convolutional architectures that directly regress parameterizations of 3D deformation fields from imaging. While some of these methods can be trained in a self-supervised fashion, they require longitudinal datasets with multiple CT volumes for every patient and/or procedure, which is infeasible for many clinical and surgical settings.

**Marker-based multi-component tracking.** Unlike the registration methods described above, some animal biomechanics studies use implanted fiducial markers to track and study the motion of bony structures in X-ray videos [43, 44]. However, this technique is impractical in clinical settings due to the invasive nature of implanting markers, as well as its inability to track deformable soft tissue.

## 3   Methods

Let $L_c^\infty(\mathbb{R}^k)$ define the set of bounded and compact functions $g : \mathbb{R}^k \to \mathbb{R}$ and $\mathbf{V} \in L_c^\infty(\mathbb{R}^3)$ represent a 3D CT volume of a patient. Additionally, let $\mathbf{I} = \{\mathbf{I}_n \in L_c^\infty(\mathbb{R}^2)\}_{n=1}^N$ represent a set of $N$ 2D X-ray images of the same patient at a different time point (we assume all images in $\mathbf{I}$ are acquired simultaneously). Specifically, assume the patient is in different positions for the acquisitions of $\mathbf{V}$ and $\mathbf{I}$ (e.g., supine vs. standing).

The geometry underlying X-ray image formation can be modeled using a pinhole camera [45]. Let each image $\mathbf{I}_n$ be associated with a camera matrix $\mathbf{\Pi}_n = \mathbf{K}_n [\mathbf{R}_n \mid \mathbf{t}_n]$, where $\mathbf{K}_n$ and $[\mathbf{R}_n \mid \mathbf{t}_n]$ are the intrinsic and extrinsic matrices, respectively. We model the relationship between $\mathbf{V}$ and $\mathbf{I}$ as

$$\mathbf{I}_n = \mathcal{P}(\mathbf{\Pi}_n) \circ \mathbf{V} \circ \mathbf{\Phi} \,, \tag{1}$$

where $\mathcal{P}(\mathbf{\Pi}_n) : L_c^\infty(\mathbb{R}^3) \to L_c^\infty(\mathbb{R}^2)$ is the X-ray projection operator whose geometry is defined by the camera matrix $\mathbf{\Pi}_n$, and $\mathbf{\Phi} : \mathbb{R}^3 \to \mathbb{R}^3$ is a 3D deformation field. Given $\mathbf{V}$ and $\mathbf{I}$, our goal is to solve for the camera matrices $\{\mathbf{\Pi}_1, \dots, \mathbf{\Pi}_N\}$ and the deformation field $\mathbf{\Phi}$.

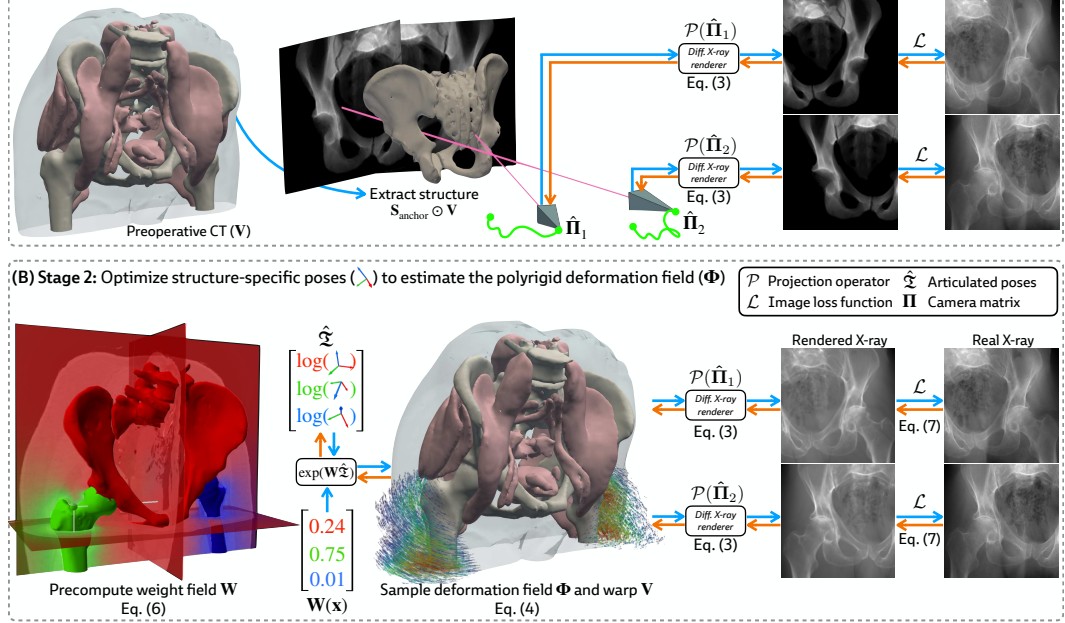

Figure 3: **Overview of PolyPose. (A)** We estimate the camera pose $\hat{\mathbf{\Pi}}$ for each X-ray by registering the structure $\mathbf{S}_{\text{anchor}}$ across all input views (Section 3.2). **(B)** Using these camera matrices, we jointly optimize the poses of the rigid bodies in $\mathbf{V}$ by producing a locally linear polyrigid warp field and maximizing the similarity of warped differentiably rendered and real X-rays (Section 3.3).

## 3.1 Preliminaries

**Differentiable X-ray rendering.** Given the camera matrix $\mathbf{\Pi}_n = \mathbf{K}_n\left[\mathbf{R}_n \mid \mathbf{t}_n\right] \in \mathbb{R}^{3 \times 4}$, the location of the X-ray source in world coordinates is given by $\mathbf{S} = -\mathbf{R}_n^T \mathbf{t}_n$ [46, p. 158]. For a pixel in $\mathbf{I}_n$ with coordinates $\mathbf{p} \in \mathbb{R}^2$, its location on the X-ray detector plane is given by $\mathbf{P} = f\mathbf{\Pi}_n^\dagger \tilde{\mathbf{p}}$, where $f$ is the X-ray machine's focal length (derived from $\mathbf{K}_n$ [46, p. 162]), $\dagger$ is the pseudoinverse, and $\tilde{\mathbf{p}} \in \mathbb{P}^2$ is $\mathbf{p}$ in homogeneous coordinates. A construction of the intrinsic matrix $\mathbf{K}_n$ is given in Appendix A.

The 3D ray back-projected from $\mathbf{p}$ to the camera center can be parameterized as $\vec{\mathbf{r}}(\lambda) = \mathbf{S} + \lambda(\mathbf{P} - \mathbf{S})$ for all $\lambda \in [0, 1]$. The negative log-intensity measured at $\mathbf{p}$ is given by the Beer-Lambert law [47]:

$$\mathbf{I}_n(\mathbf{p}) = \int_{\mathbf{x} \in \vec{\mathbf{r}}} \mathbf{V}(\mathbf{x})\mathrm{d}\mathbf{x} = \int_0^1 \mathbf{V}\big(\vec{\mathbf{r}}(\lambda)\big)\|\vec{\mathbf{r}}'(\lambda)\|\mathrm{d}\lambda = \|\mathbf{P} - \mathbf{S}\| \int_0^1 \mathbf{V}\big(\mathbf{S} + \lambda(\mathbf{P} - \mathbf{S})\big)\mathrm{d}\lambda, \quad (2)$$

where $\mathbf{V}(\cdot)$ represents the linear attenuation coefficient (LAC) at every point in space, a physical property proportional to the density. The line integral in Eq. (2) defines the first-order continuous approximation of the X-ray projection operator $\mathcal{P}(\mathbf{\Pi}_n)$, i.e., no scattering, beam hardening, etc.

We implement Eq. (2) by modeling $\mathbf{V}$ with a discrete CT volume (i.e., a voxelgrid of LACs). This discrete line integral can be approximated with interpolating quadrature as

$$\mathbf{I}_n(\mathbf{p}) \approx \|\mathbf{P} - \mathbf{S}\| \sum_{m=1}^{M-1} \mathbf{V}\left[\mathbf{S} + \lambda_m(\mathbf{P} - \mathbf{S})\right](\lambda_{m+1} - \lambda_m), \quad (3)$$

where $\lambda_{m+1} - \lambda_m$ is the distance between adjacent samples on $\vec{\mathbf{r}}$ and $\mathbf{V}[\cdot]$ represents a sampling operation (e.g., trilinear interpolation) on the discrete volume [48, 49]. Here, we rely on open-source implementations of the rendering equation (3) as a series of vectorized tensor operations [50].

**Parameterizing the deformation field.** Let $\{\mathbf{S}_1, \ldots, \mathbf{S}_K\} \subset \mathbf{V}$ represent a set of disjoint binary masks for the articulated rigid bodies within the volume (e.g., the bones of the skeleton). Each structure $\mathbf{S}_k$ is associated with a corresponding rigid transformation $\mathbf{T}_k \in \mathbf{SE}(3)$ that represents the displacement of $\mathbf{S}_k$ between the acquisitions of $\mathbf{V}$ and $\mathbf{I}$. In the polyrigid framework, the deformation

field $\boldsymbol{\Phi}$ is parameterized as a convex combination of the $K$ rigid transforms represented in the tangent space $\mathfrak{se}(3)$ [26]. Specifically, the polyrigid deformation at any point $\mathbf{x} \in \mathbb{R}^3$ is computed as

$$\boldsymbol{\Phi}[\mathbf{T}_1, \ldots, \mathbf{T}_K](\mathbf{x}) = \overline{\mathbf{T}}(\mathbf{x})\tilde{\mathbf{x}}, \quad \text{where} \quad \overline{\mathbf{T}}(\mathbf{x}) \triangleq \exp\left(\frac{\sum_{k=1}^{K} w_k(\mathbf{x}) \log \mathbf{T}_k}{\sum_{k=1}^{K} w_k(\mathbf{x})}\right) \in \mathbf{SE}(3) \quad (4)$$

is the locally-rigid transformation at $\mathbf{x}$ (represented as a $4 \times 4$ matrix), $\tilde{\mathbf{x}} \in \mathbb{P}^3$ is the representation of $\mathbf{x} \in \mathbb{R}^3$ in homogeneous coordinates, $w_k(\mathbf{x})$ is the weight of structure $\mathbf{S}_k$ at $\mathbf{x}$, and $\log(\cdot)$ and $\exp(\cdot)$ are the logarithm and exponential maps for $\mathbf{SE}(3)$, respectively.

By fusing log-transformed versions of the pose for each structure, as opposed to simply averaging their associated displacements, the resulting polyrigid warp is diffeomorphic, anatomically constrained, and well-suited to our ill-posed setting. Eq. (4) can also be efficiently computed using closed forms for $\log(\cdot)$ and $\exp(\cdot)$ maps on $\mathbf{SE}(3)$, which are provided in Appendix B.

## 3.2 Estimating the Camera Matrices

Given a preoperative 3D volume $\mathbf{V}$ and intraoperative 2D X-ray images $\mathbf{I}_1, \ldots, \mathbf{I}_N$, we aim to estimate the camera matrices $\boldsymbol{\Pi}_1, \ldots, \boldsymbol{\Pi}_N$. While patients move non-rigidly between the acquisitions of $\mathbf{V}$ and $\mathbf{I}$, there exists a global rigid transform for an *individual articulated structure*. Therefore, using a rigid 2D/3D registration framework (xvr [16]), we anchor pose representations by first rigidly aligning a structure $\mathbf{S}_{\text{anchor}}$ that is reliably visible across all views in $\mathbf{I}$, such as the pelvis in Figure 3A. Using $\mathbf{S}_{\text{anchor}}$, we estimate the extrinsic matrix for every X-ray image $[\hat{\mathbf{R}}_n \mid \hat{\mathbf{t}}_n]$. Finally, as X-ray imaging systems used in clinical practice are calibrated, the intrinsic parameters $\mathbf{K}_1, \ldots, \mathbf{K}_N$ can easily be obtained from each image's metadata, yielding camera matrices $\hat{\boldsymbol{\Pi}}_n = \mathbf{K}_n[\hat{\mathbf{R}}_n \mid \hat{\mathbf{t}}_n]$.

## 3.3 Constructing the Polyrigid Deformation Field

**Constructing the weight field.** Prior formulations of 3D/3D polyrigid registration [51] have proposed defining the weight of each structure $\mathbf{S}_k$ at any point $\mathbf{x} \in \mathbb{R}^3$ using the reciprocal distance function

$$w_k(\mathbf{x}) = \frac{1}{1 + \epsilon d_k^2(\mathbf{x})}, \quad (5)$$

where $d_k$ is the minimum Euclidean distance from $\mathbf{x}$ to $\mathbf{S}_k$, and $\epsilon \leq 1$ is a hyperparameter controlling the rate of decay of $w_k$ as $\mathbf{x}$ moves further away from $\mathbf{S}_k$. However, we found that Eq. (5) produced inaccurate deformation fields for volumes containing articulated bodies with very different sizes (Table 3). To our knowledge, Eq. (5) has largely only been used when the constituent substructures have comparable volumes, such as certain brain regions [51] or the carpal bones [52, 53].

Instead, loosely inspired by the influence of mass in gravitational attraction [54], we define the weight field for each structure as

$$w_k(\mathbf{x}) = \frac{m_k}{1 + d_k^2(\mathbf{x})}, \quad (6)$$

where $m_k$ is the normalized mass of $\mathbf{S}_k$ relative to all structures. We estimate $m_k$ using the volume of $\mathbf{S}_k$ (i.e., assuming a constant density for all bones). This formulation eliminates challenging hyperparameter optimization while still producing topologically valid deformations (Table 3). An example of our proposed weight field is visualized in Figure 3B (*left*).

**Joint optimization.** Given the camera matrices $\hat{\boldsymbol{\Pi}}_1, \ldots, \hat{\boldsymbol{\Pi}}_N$ estimated in Section 3.2, we jointly optimize the pose for every rigid body by maximizing an image similarity metric $\mathcal{L}$ (e.g., normalized cross correlation, mutual information, etc.) between the rendered and real X-ray images:

$$(\hat{\mathbf{T}}_1, \ldots, \hat{\mathbf{T}}_K) = \underset{\mathbf{T}_1, \ldots, \mathbf{T}_K}{\arg\max} \frac{1}{N} \sum_{n=1}^{N} \mathcal{L}\left(\mathbf{I}_n, \mathcal{P}(\hat{\boldsymbol{\Pi}}_n) \circ \mathbf{V} \circ \boldsymbol{\Phi}[\mathbf{T}_1, \ldots, \mathbf{T}_K]\right), \quad (7)$$

where $\boldsymbol{\Phi}$ is constructed from $\mathbf{T}_1, \ldots, \mathbf{T}_K$ via Eq. (4).

**Efficient computation with a vectorized forward model.** Let $\mathbf{X} \in \mathbb{R}^{M \times 3}$ be the coordinates of every voxel in $\mathbf{V}$ where $M$ is the number of voxels. For each structure $\mathbf{S}_k$, we evaluate Eq. (6) to precompute $\mathbf{w}_k(\mathbf{x})$ at every $\mathbf{x} \in \mathbf{X}$. Concatenating the structure-specific weights, we construct the

discretized weight field $\mathbf{W} \in \mathbb{R}^{M \times K}$, with its rows normalized to sum to 1. Additionally, since the codomain of the logarithm map $\log : \mathbf{SE}(3) \to \mathfrak{se}(3)$ is homeomorphic to $\mathbb{R}^6$ (see Appendix B), we succinctly represent all structure-specific transformations $\hat{\mathbf{T}}_1, \ldots, \hat{\mathbf{T}}_K$ with the matrix

$$\hat{\boldsymbol{\mathfrak{T}}} = \begin{bmatrix} \text{---}\log \hat{\mathbf{T}}_1 \text{---} \\ \vdots \\ \text{---}\log \hat{\mathbf{T}}_K \text{---} \end{bmatrix} \in \mathbb{R}^{K \times 6}. \tag{8}$$

Then, using batched matrix multiplication, we construct the polyrigid warp at all voxel coordinates:

$$\hat{\boldsymbol{\Phi}}(\mathbf{X}) = \exp(\mathbf{W}\hat{\boldsymbol{\mathfrak{T}}})\tilde{\mathbf{X}} \in \mathbb{R}^{M \times 3}, \tag{9}$$

where $\exp(\mathbf{W}\hat{\boldsymbol{\mathfrak{T}}}) \subset \mathbf{SE}(3)$ represents a set of $M$ rigid transforms computed with a vectorized implementation of the exponential map. The computational flow in PolyPose is illustrated in Figure 3B.

### 3.4 Implementation Details

To measure the similarity between rendered and real X-rays ($\mathcal{L}$ in Figure 3), we use a variant of the patch-wise normalized cross correlation loss [55] that computes the similarity between raw and Sobel-filtered images at multiple scales [15, 56]. For both camera and structure-specific pose estimation, we perform gradient-based optimization on rigid transforms parameterized in the tangent space $\mathfrak{se}(3)$. Across all experiments, we use the Adam optimizer [57] with step sizes $\beta_{\mathrm{rot}} = 10^{-2}$ and $\beta_{\mathrm{xyz}} = 10^0$ for the rotational and translational components of $\mathfrak{se}(3)$, respectively. PolyPose and all baseline methods were trained (if applicable) and evaluated using a single NVIDIA RTX A6000. All further implementation details are provided in Appendix C.

## 4 Experiments

### 4.1 Datasets and Experimental Setup

**Head&Neck.** We first perform experiments on a longitudinal dataset of CT scans of 31 patients undergoing radiotherapy for head and neck squamous cell carcinoma [58] using a 10/2/19 subject-wise training, validation, and testing split. Each patient had one CT volume from the pre-, peri-, and post-treatment periods, respectively [59]. To simulate a deformable 2D/3D registration task, we generated a small set of synthetic X-ray images (2-9 images) in a $180°$ orbit from either the peri- or post-treatment CTs (fixed image) to be registered to the preoperative CT (moving image). To assess registration accuracy, we measure the 3D volume overlap between the warped labelmaps of rigid and soft tissue structures and their corresponding ground truth labelmaps in the peri- or post-treatment CT. The poses of soft tissue structures are not optimized, thereby serving to assess PolyPose's extrapolation outside rigid bodies.

**DeepFluoro.** To measure performance on real X-ray images, we use DeepFluoro, a cadaveric orthopedic surgery dataset of six preoperative CT volumes with associated intraoperative X-ray images (between 24-111 per subject) [60]. As is typical in image-guided interventions, the intraoperative X-ray images were acquired from a limited viewing angle (approximately $30°$) as unconventional oblique views are often not useful for human operators. Additionally, DeepFluoro provides manual segmentations of the femurs and pelvis in the real X-ray images. As such, for each subject, we estimate a deformation field using two X-rays capturing the left and right femurs, and quantitatively evaluate registration accuracy using 2D segmentation metrics computed on X-ray images not used to estimate the deformation field.

### 4.2 Baselines

We evaluate several 2D/3D and 3D/3D registration approaches as points of reference with implementation details provided in Appendix D. We first compare against xvr [16], the current state-of-the-art method for estimating a single global rigid transformation. Next, we evaluate two convolutional deep learning methods for deformable 2D/3D registration: LiftReg [18] and 2D3D-RegNet [19]. LiftReg regresses the coefficients for a low-rank approximation of the deformation field whose basis is obtained via PCA on a training set of ground truth 3D/3D warps, while 2D3D-RegNet directly estimates a dense translation field using a VoxelMorph-style approach [61].

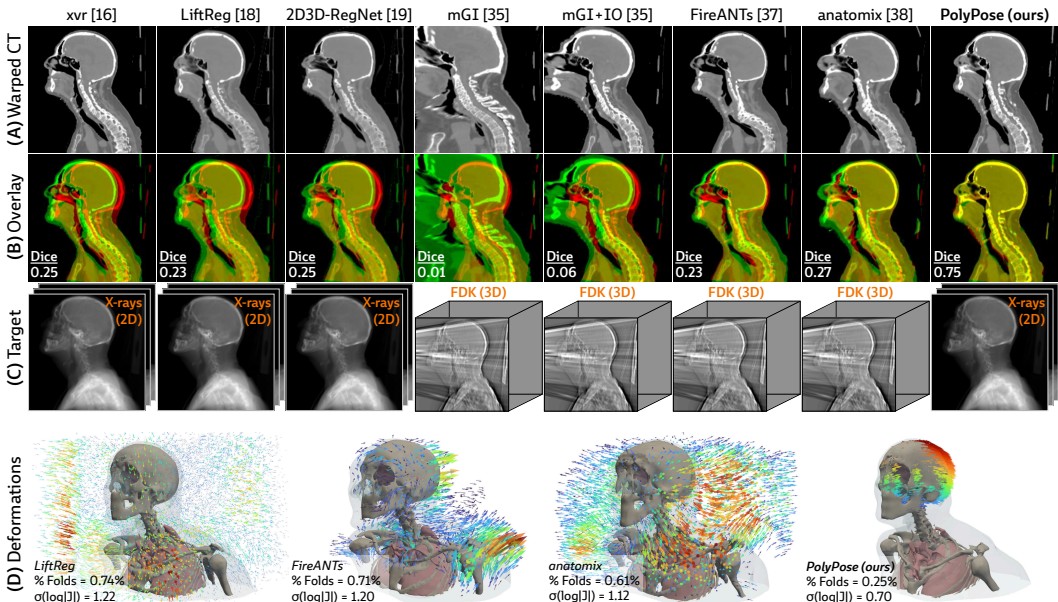

Figure 4: **Qualitative evaluations of sparse-view 2D/3D registration on Head&Neck. (A)** Resulting warped CT volumes by different registration methods. **(B)** We visualize registration error by overlaying the warped CT (green) on the ground truth CT (red). Baseline methods incur registration errors in the skull, spine, and surrounding soft tissue. **(C)** 2D/3D registration methods take stacks of X-ray images as input, while 3D/3D registration methods require a reconstructed volume. **(D)** Visualizations of the estimated deformation fields, superimposed on renderings of the warped CT volumes. PolyPose estimates smooth, localized deformations with minimal topological errors. Visualizations of the deformation fields for all other baselines are provided in Appendix E.1.

As 3D volumes can be rapidly reconstructed from intraoperative 2D X-rays to serve as registration targets, we also compare PolyPose to four 3D/3D registration methods [34, 35, 37, 38]. To match the speed requirements of intraoperative settings, we reconstruct 3D volumes using the FDK algorithm [62] implemented in the ASTRA Toolbox [33]. Both uniGradICON (uGI) [34] and multiGradICON (mGI) [35], a pair of foundation models for unimodal and multimodal image registration, contain variants with *post-hoc* iterative optimization (+IO). For each experiment, we report the two best-performing variants from uGI, uGI+IO, mGI, and mGI+IO. FireANTs [37] and anatomix [38] are iterative solvers that provide state-of-the-art 3D/3D registration via improved optimization techniques and feature representations, respectively.

## 4.3 Results

**Sparse-view registration.** Figure 4 visualizes the warped CT volumes and deformation fields estimated from three input views distributed across a $180°$ viewing angle range and Figure 5 reports quantitative evaluation metrics for the Head&Neck dataset. Of all evaluated methods, PolyPose estimates the most accurate deformation fields across all numbers of input X-rays available as registration targets. PolyPose achieves the highest 3D Dice on both rigid structures and important soft tissue organs, even though the pose of these organs was not directly estimated during optimization. This is crucial as non-target organs are to be avoided as much as possible in the delivery of radiotherapy. Of particular note, PolyPose outperforms both deep learning-based 2D/3D methods [18, 19], suggesting that training on the limited datasets available in interventional settings produces models that fail to generalize.

PolyPose also estimates deformation fields with minimal topological defects. Our construction from a small number of rigid components yields interpretable deformation fields that are more anatomically plausible than baselines. For example, in a subject with only minimal head motion, PolyPose recovers the exact underlying deformation (Figure 4D), whereas anatomix [38], the second-most accurate method, produces topologically-defective and irregular warps as measured by the percentage of folds in the deformation, %Folds, and the standard deviation of volume changes, $\sigma(\log|\mathbf{J}|)$ [63].

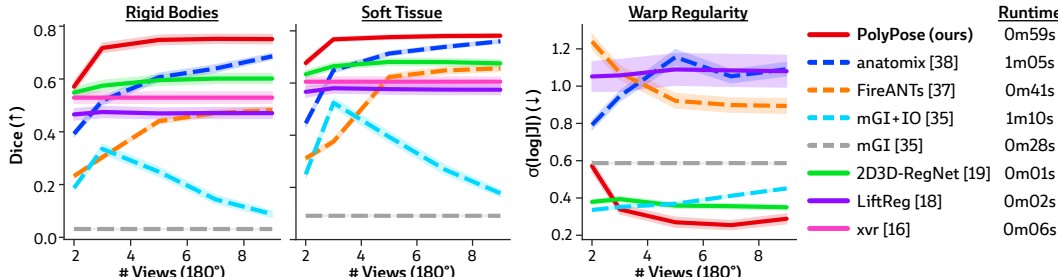

Figure 5: **Quantitative results of sparse-view 2D/3D registration on the Head&Neck dataset.** We evaluated the accuracy of estimated deformation fields by computing the 3D Dice on 21 rigid structures (L/R humerus, L/R scapula, L/R clavicles, thoracic and cervical vertebrae, and skull) and five soft tissue structures (thyroid, spinal cord, brain, esophagus, and trachea). PolyPose is the most accurate registration method and also exhibits the most regular deformable warps for almost all numbers of views. 2D/3D and 3D/3D registration methods are shown with solid and dashed lines, respectively. Lastly, we report the average runtime for each method.

**Limited-angle registration.** Certain baselines are not applicable to the DeepFluoro dataset. The deep learning methods LiftReg [18] and 2D3D-RegNet [19] cannot be trained on this dataset since they require multiple CTs from each patient, while each subject in DeepFluoro only has a single volume. Therefore, we also evaluate a regularized dense deformation model from radiotherapy, which optimizes a displacement vector for every voxel [23]. In Figure 3, we visualize a preoperative CT and two intraoperative X-rays spaced about $30°$ apart and the deformation field estimated by PolyPose.

Visualizations of the estimated deformation fields and warped CTs show that PolyPose produces interpretable warps, e.g., modeling the external rotation of the femurs (Figure 6A and B). In contrast, the dense parameterizations yield anatomically implausible and unintelligible deformations as their objective prioritizes memorizing the appearance of the training views. Additionally, dense deformation models can only influence voxels on which they have direct pixel supervision (see the broken femurs in Figure 6B), whereas PolyPose extrapolates to unsupervised anatomy via piecewise-rigidity.

To measure the accuracy of the estimated deformation fields, we warp the input CTs, render synthetic X-rays from them, and compare the positions of bones in the rendered X-rays with their ground truth segmentations in the real X-rays (Figure 6D). Table 1 reports the 2D Dice and 95th percentile Hausdorff Distance (HD95) for the pelvis, left femur, and right femur, as well as the %Folds in the estimated deformation fields. We used the pelvis as the anchor when estimating the camera poses for the X-ray images (Figure 3A). As such, nearly all baselines (evaluated using our camera matrices) exhibit high accuracy on the pelvis. However, for the femurs, PolyPose produces the highest accuracy. Additional visualizations of all baselines in Table 1 are provided in Appendix E.2.

### 4.4 Ablations and Analyses

**Choice of deformation parameterization.** In Table 2, we compare our polyrigid formulation to per-voxel translations [23] and $\mathbf{SE}(3)$ transformations [32, 42], also optimized via differentiable rendering. Given minimal supervision, only our low-dimensional deformation model enables the localization of

Table 1: **Quantitative results on limited-angle registration with the DeepFluoro dataset.** Given only two X-ray images with $30°$ of separation, PolyPose recovers the most accurate 3D deformation field relative to all baselines, while also having no topological defects. We color the **best** and second-best methods and report all metrics as *mean(sd)*.

| | Pelvis | | Femur (L) | | Femur (R) | | |
|---|---|---|---|---|---|---|---|
| | Dice ($\uparrow$) | HD95 ($\downarrow$) | Dice ($\uparrow$) | HD95 ($\downarrow$) | Dice ($\uparrow$) | HD95 ($\downarrow$) | % Folds ($\downarrow$) |
| **PolyPose (ours)** | **0.99(0.00)** | **1.00(0.00)** | **0.98(0.01)** | **1.48(1.09)** | **0.98(0.01)** | **1.56(1.02)** | **0.00(0.00)%** |
| Dense $\mathbb{R}^3$ [23] | 0.98(0.01) | 3.60(5.47) | 0.97(0.02) | 3.29(2.62) | 0.96(0.02) | 3.43(2.78) | 0.44(0.12)% |
| xvr [16] | **0.99(0.00)** | 1.01(0.07) | 0.96(0.02) | 4.03(3.07) | 0.94(0.02) | 6.51(4.21) | **0.00(0.00)%** |
| FireANTs [37] | **0.99(0.00)** | 1.01(0.07) | 0.96(0.02) | 4.03(3.07) | 0.93(0.02) | 9.63(4.26) | **0.00(0.00)%** |
| anatomix [38] | 0.95(0.01) | 3.63(0.50) | 0.93(0.02) | 5.44(2.77) | 0.92(0.2) | 6.89(4.13) | 3.01(1.21)% |
| multiGradICON [35] | 0.83(0.05) | 16.37(6.75) | 0.86(0.04) | 8.69(4.84) | 0.77(0.08) | 15.18(3.54) | **0.00(0.00)%** |
| uniGradICON [34] | 0.66(0.07) | 21.98(4.57) | 0.50(0.12) | 28.51(12.71) | 0.83(0.04) | 13.74(0.98) | **0.00(0.00)%** |

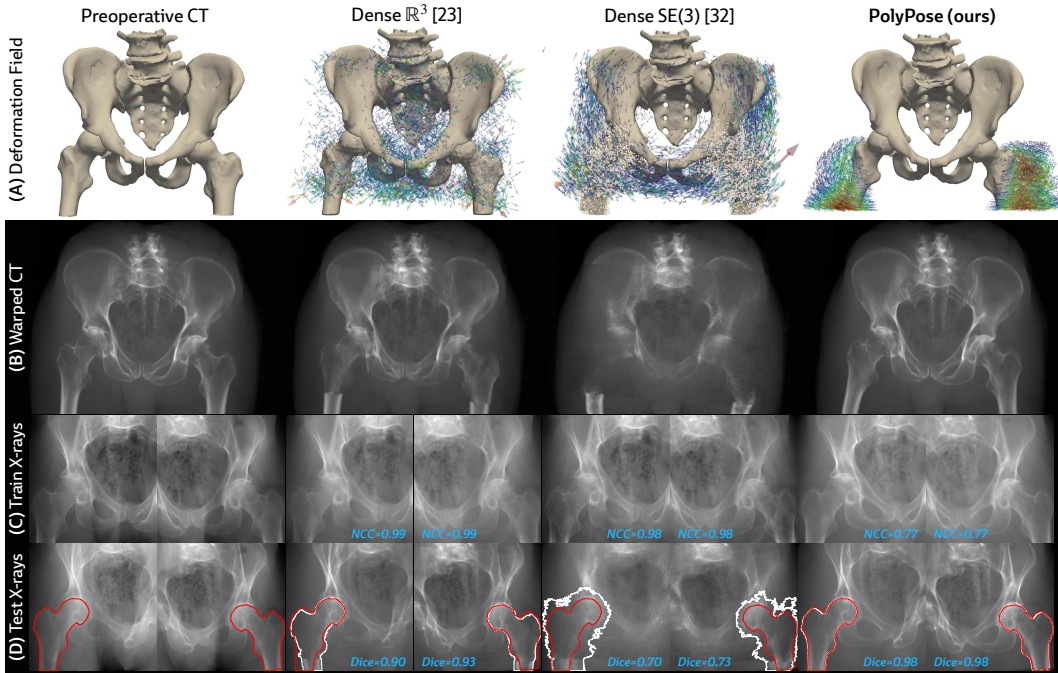

Figure 6: **Qualitative evaluations of limited-angle 2D/3D registration on DeepFluoro.** (**A** and **B**) PolyPose's anatomical priors recover realistic motion even without direct supervision. (**C**) Dense deformation parameterizations [23, 32] estimate warps that reproduce the appearance of the training X-rays, yielding near-perfect image similarity metrics (NCC $\approx 0.99/1$). (**D**) However, the dense deformations do not generalize to held-out views, as demonstrated by the misalignment of the ground truth (red) and estimated (white) segmentation labels in unseen images.

the misaligned femurs without topological defects. PolyPose has only $\mathcal{O}(K)$ optimizable parameters and is thus well suited for ill-posed settings, whereas the under-constrained dense representations have $\mathcal{O}(M)$ parameters with $K \ll M$. Here, $K = 3$ and $M = 398 \times 197 \times 398 \approx 10^7$.

**Choice of weight function.** In Table 3, we compare different parameterizations of the weight field. Our hyperparameter-free weighting function in Eq. (6) outperforms the widely used formulation in Eq. (5). Note that the optimal performance for the left and right femurs is achieved for vastly different hyperparameter values ($\epsilon = 10^0$ vs. $\epsilon = 10^{-3}$) when using Eq. (5). Thus, Eq. (5) has a large hyperparameter search space, requiring a different $\epsilon$ for every rigid body. In contrast, our hyperparameter-free function in Eq. (6) uses the mass of each rigid body as an effective heuristic.

**Number of rigid components.** PolyPose is memory-efficient, capable of jointly optimizing the poses of 26 rigid bodies in a large CT scan on a single GPU with 48 GB of vRAM. However, this may be too computationally expensive for resource-limited medical settings. We therefore perform an ablation on the Head&Neck dataset where we systematically increase the number of structures whose poses we optimize. Starting from rigid pre-alignment (i.e., without PolyPose), we progressively add structures until reaching the full configuration used in Figure 5. Including the skull, cervical spine, and thoracic spine stabilizes the deformation fields estimated by PolyPose (Figure 7A). Adding further rigid bodies yields only marginal improvements, demonstrating that PolyPose is expressive even when constrained to a subset of rigid bodies in an anatomical region.

Table 2: **Performance of different deformation parameterizations on DeepFluoro.** PolyPose successfully recovers the position of the femurs, while the dense representations fail to do so.

| | Pelvis | | Femur (L) | | Femur (R) | | |
|---|---|---|---|---|---|---|---|
| | Dice ($\uparrow$) | HD95 ($\downarrow$) | Dice ($\uparrow$) | HD95 ($\downarrow$) | Dice ($\uparrow$) | HD95 ($\downarrow$) | % Folds ($\downarrow$) |
| **PolyPose (ours)** | **0.99(0.00)** | **1.00(0.00)** | **0.98(0.01)** | **1.48(1.09)** | **0.98(0.01)** | **1.56(1.02)** | **0.00(0.00)%** |
| Dense $\mathbb{R}^3$ [23] | 0.98(0.01) | 3.60(5.47) | 0.97(0.02) | 3.29(2.62) | 0.96(0.02) | 3.43(2.78) | 0.44(0.12)% |
| Dense $\mathbf{SE}(3)$ [32] | 0.93(0.02) | 9.42(5.69) | 0.90(0.02) | 6.07(2.01) | 0.88(0.03) | 9.29(3.41) | 44.08(00.00)% |

Table 3: **Performance of different weight functions on DeepFluoro.** Our hyperparameter-free weighting function (6) outperforms the previously proposed Eq. (5), which achieves optimal performance for various anatomical structures at different hypermeter values.

| Weight function | $\epsilon$ | Pelvis Dice ($\uparrow$) | HD95 ($\downarrow$) | Femur (L) Dice ($\uparrow$) | HD95 ($\downarrow$) | Femur (R) Dice ($\uparrow$) | HD95 ($\downarrow$) | % Folds ($\downarrow$) |
|---|---|---|---|---|---|---|---|---|
| **Eq. (6)** | – | **0.99(0.00)** | **1.00(0.00)** | **0.98(0.01)** | **1.48(1.09)** | **0.98(0.01)** | **1.56(1.02)** | **0.00(0.00)%** |
| Eq. (5) | $10^0$ | **0.99(0.00)** | 1.38(0.41) | 0.93(0.02) | 5.60(3.29) | 0.96(0.01) | 3.29(3.48) | 0.03(0.01)% |
| Eq. (5) | $10^{-1}$ | **0.99(0.00)** | 1.58(0.41) | 0.93(0.02) | 5.31(3.27) | 0.96(0.01) | 3.53(3.55) | 0.02(0.01)% |
| Eq. (5) | $10^{-2}$ | **0.99(0.00)** | 1.49(0.37) | 0.94(0.01) | 4.24(2.45) | 0.95(0.01) | 4.27(3.75) | **0.00(0.00)%** |
| Eq. (5) | $10^{-3}$ | 0.98(0.00) | 1.62(0.36) | 0.95(0.01) | 2.87(1.18) | 0.95(0.01) | 4.34(3.71) | **0.00(0.01)%** |

**Robustness to label corruption.** PolyPose requires segmentations of the relevant rigid bodies in a CT scan to construct the weight field. While there exist numerous (semi-)automated tools that make CT segmentation a relatively straightforward task [64–66], they can make notable errors and miss fine details. We therefore analyze PolyPose's performance as a function of segmentation accuracy. To simulate typical annotation errors, we systematically corrupt the ground truth segmentations in the DeepFluoro dataset with increasing radii of erosion (Figure 7B). A radius of 0 mm corresponds to no erosion and is equivalent to the experimental setting in Table 1. We find that PolyPose is robust to extreme segmentation corruption, even outperforming the baselines in Table 1 over a range of erosion radii from 1 mm to 3 mm, both in terms of Dice and Hausdorff Distance.

## 5  Discussion

**Limitations and future work.** PolyPose's ability to generalize to *extreme* deformations in soft tissue far away from skeletal structures remains to be fully evaluated. Preliminary experiments in Appendix E.3 show that PolyPose successfully models free-breathing respiratory motion between maximum inhalation and maximum expiration acquisitions. However, PolyPose may fail to generalize to settings where there are insufficient rigid bodies to constrain the deformation of soft tissues (e.g., within the abdomen). Additionally, while our method produces diffeomorphisms by construction (typically a highly desirable property), this does not cover every type of deformation. For example, separating a rigid body into two (e.g., opening the jaw) cannot be represented by a diffeomorphism and thus cannot be modeled by PolyPose. We visualize such a failure case in Appendix F. This limitation could be mitigated by the incorporation of a kinematic chain into the rigid body parameterization.

**Conclusion.** Deformable 2D/3D registration holds immense promise in localizing critical organs from intraoperative images. However, the accuracy of previous methods fails to meet the standards for clinical deployment. We present PolyPose, an optimization-based method that solves this extremely under-determined registration problem with a polyrigid field. Throughout extensive experiments on publicly available datasets from diverse clinical specialties, PolyPose estimated the most accurate and topologically correct warps in both sparse-view and limited-angle settings. In addition to its high performance, PolyPose's lack of need for regularization and near-absence of hyperparameters make it generically applicable across a broad set of medical procedures.

**(A) Ablation on number of rigid components.**

| Structures | Dice ($\uparrow$) Rigid Bodies | Soft Tissues |
|---|---|---|
| Rigid pre-alignment [16] | 0.51 | 0.49 |
| + Skull | 0.61 | 0.63 |
| + C-spine | 0.64 | 0.76 |
| + T-spine | 0.70 | 0.77 |
| + Humerus (L/R) | 0.70 | 0.81 |
| + Scapula (L/R) | 0.71 | 0.80 |
| + Clavicles (R) | 0.74 | 0.81 |

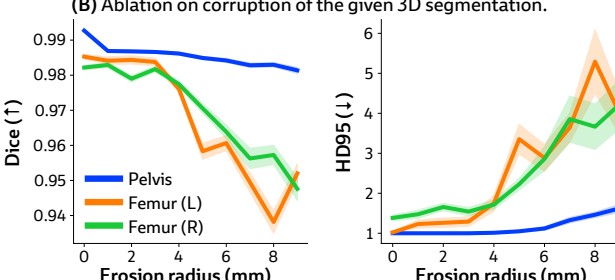

Figure 7: **PolyPose is robust to label restriction and corruption. (A)** PolyPose remains expressive even when optimizing a limited subset of the rigid bodies in an anatomical region. **(B)** PolyPose's performance is relatively stable up to 3 mm of label erosion (corresponding to a 40-60% volume reduction), beyond which registration accuracy degrades gracefully.

## Acknowledgments and Disclosure of Funding

This work was supported by NIH NIBIB 5T32EB001680-19, NIH NIBIB R01EB033773, and the Chou Family Transformative Research Fund.

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

## A  Projective X-ray Geometry

To complete the derivation of the forward model for the negative log-intensity at a pixel $\mathbf{p}$ in an X-ray image, we must specify how to construct the intrinsic matrix $\mathbf{K}$ from the image's metadata.

The intrinsic matrix represents the mapping from camera to pixel coordinates [46]. This can be decomposed as a first mapping from camera to image coordinates and a second mapping from image to pixel coordinates:

$$\mathbf{K} = \begin{bmatrix} 1/s_x & 0 & W/2 \\ 0 & 1/s_y & H/2 \\ 0 & 0 & 1 \end{bmatrix} \begin{bmatrix} f & 0 & o_x \\ 0 & f & o_y \\ 0 & 0 & 1 \end{bmatrix}, \tag{10}$$

where $f$ is the camera's focal length, $(o_x, o_y)$ is the camera's optical center, $(s_x, s_y)$ is the pixel spacing, and $(H, W)$ is the image's height and width, respectively [16].

These intrinsic parameters for each X-ray image can readily be identified from the image's metadata encoded in the DICOM (Digital Imaging and Communications in Medicine) header. Specifically,

- The focal length $f$ is given by the `DistanceSourceToDetector` (0018,1110) attribute.
- The optical center $(o_x, o_y)$ is given by the `DetectorActiveOrigin` (0018,7028) attribute.
- The pixel spacing $(s_x, s_y)$ is the given by the `ImagerPixelSpacing` (0018,1164) attribute.
- The image dimensions $(H, W)$ are given by the `Rows` (0028,0010) and `Columns` (0028,0011) attributes, respectively.

## B  Lie Theory for Polyrigid Transforms

We summarize the Lie theory of $\mathbf{SE}(3)$ from Blanco [67] needed to implement PolyPose. We start by defining the logarithmic map, which maps any rigid transformation

$$\mathbf{T} = \begin{bmatrix} \mathbf{R} & \mathbf{t} \\ \mathbf{0} & 1 \end{bmatrix} \in \mathbf{SE}(3), \quad \text{where} \quad \mathbf{R} \in \mathbf{SO}(3) \text{ and } \mathbf{t} \in \mathbb{R}^3, \tag{11}$$

to the vector $\mathbf{v} = \begin{bmatrix} \boldsymbol{\omega} & \boldsymbol{u} \end{bmatrix}^T \in \mathfrak{se}(3) \cong \mathbb{R}^6$. This vector corresponds to the matrix

$$\log(\mathbf{T}) \triangleq \begin{bmatrix} 0 & -\omega_3 & \omega_2 & u_1 \\ \omega_3 & 0 & -\omega_1 & u_2 \\ -\omega_2 & \omega_1 & 0 & u_3 \\ 0 & 0 & 0 & 0 \end{bmatrix}, \tag{12}$$

which itself is the generator of an infinitesimal rototranslation about the axis defined by $\boldsymbol{u}$.

To efficiently write the formulas for $\boldsymbol{\omega}$ and $\boldsymbol{u}$, it is convenient to first define the exponential map:

$$\exp(\mathbf{v}) = \begin{bmatrix} e^{[\boldsymbol{\omega}]_\times} & \boldsymbol{\Omega u} \\ \mathbf{0} & 1 \end{bmatrix}, \tag{13}$$

where

$$e^{[\boldsymbol{\omega}]_\times} = \mathbf{I} + \frac{\sin\theta}{\theta}[\boldsymbol{\omega}]_\times + \frac{\theta - \cos\theta}{\theta^2}[\boldsymbol{\omega}]_\times^2, \tag{14}$$

$$\boldsymbol{\Omega} = \mathbf{I} + \frac{1 - \cos\theta}{\theta^2}[\boldsymbol{\omega}]_\times + \frac{\theta - \sin\theta}{\theta^3}[\boldsymbol{\omega}]_\times^2, \tag{15}$$

and $\theta = \|\boldsymbol{\omega}\|$ and $[\cdot]_\times$ constructs a skew-symmetric matrix from a 3-vector.

Then, $\boldsymbol{\omega}$ is given by Rodrigues' rotation formula

$$\boldsymbol{\omega} = \frac{1}{2\sin\theta} \begin{bmatrix} \mathbf{R}_{32} - \mathbf{R}_{23} \\ \mathbf{R}_{13} - \mathbf{R}_{31} \\ \mathbf{R}_{21} - \mathbf{R}_{12} \end{bmatrix}, \quad \text{where} \quad \theta = \arccos\left(\frac{\text{trace}(\mathbf{R}) - 1}{2}\right), \tag{16}$$

and $\boldsymbol{u} = \boldsymbol{\Omega}^{-1}\mathbf{t}$.

## C  Additional Implementation Details

### C.1  Estimating Camera Poses

To recover camera poses in an accurate and automatic manner, we use xvr, a patient-specific machine learning framework for state-of-the-art rigid 2D/3D registration [16]. Specifically, given $\mathbf{V}$, we train a patient-specific convolutional network $\mathbf{f}_\theta : \mathbf{I} \to [\mathbf{R} \mid \mathbf{t}]$ to predict an initial camera pose estimate for a given X-ray image using self-supervised synthetic pretraining. At inference time, we refine these initial pose estimates using differentiable rendering, a protocol known as test-time optimization (Figure 3A). However, we modify the original test-time optimization protocol and instead optimize the pose of a single anatomical structure $\mathbf{S}_{\text{anchor}} \in \{\mathbf{S}_1, \ldots, \mathbf{S}_K\}$. We anchor our representation of the camera poses by rigidly registering the left clavicle in the Head&Neck dataset and the pelvis in the DeepFluoro dataset.

**Optimization problem.** Given an image similarity loss function $\mathcal{L}$ (e.g., normalized cross correlation, mutual information, etc.), we estimate the extrinsic parameters of each X-ray image by independently solving the following optimization problem:

$$[\hat{\mathbf{R}}_n \mid \hat{\mathbf{t}}_n] = \operatorname*{argmax}_{[\mathbf{R}_n \mid \mathbf{t}_n] \in \mathbf{SE}(3)} \mathcal{L}\Big(\mathbf{I}_n, \mathcal{P}\big(\mathbf{K}_n[\mathbf{R}_n \mid \mathbf{t}_n]\big) \circ \big(\mathbf{S}_{\text{anchor}} \odot \mathbf{V}\big)\Big) \tag{17}$$

where $\odot$ is element-wise multiplication used to mask the CT volume and render the structure $\mathbf{S}_{\text{anchor}}$. This optimization is performed in the tangent space of $\mathbf{SE}(3)$ using gradient descent. Finally,

$$\hat{\mathbf{\Pi}}_n = \mathbf{K}_n[\hat{\mathbf{R}}_n \mid \hat{\mathbf{t}}_n] . \tag{18}$$

We use the hybrid loss function gradient multiscale normalized cross correlation (gmNCC) to guide 2D/3D rigid registration. This composite loss function is the average of multiscale NCC (mNCC) [15], which averages NCC across the global and local scales, and gradient NCC (gNCC) [60], which computes NCC on Sobel-filtered versions of the image. This image similarity metric is advantageous for 2D/3D registration tasks as mNCC encourages global alignment while gNCC encourages alignment of edges of bones.

### C.2  Polyrigid Pose Optimization

**Weight field.** Given a labelmap for the preoperative CT scan, we first compute structure-specific Euclidean distance transforms for each rigid body whose pose we will optimize. Examples of these per-structure distance fields are illustrated for a subject in the DeepFluoro dataset (Figure 8). Finally, these distance transform are combined using Eq. (6). Since the weights are fixed during optimized in PolyPose, this field can be precomputed before estimating the warp field.

**Optimization.** We represent the poses of every rigid body in the tangent space $\mathfrak{se}(3)$. Since translational parameters ($\boldsymbol{u}$) in units of millimeters are typically two orders of magnitude larger than angular parameters ($\boldsymbol{\omega}$) in units of radians, we use two separate step sizes. Specifically, we use the Adam optimizer with step sizes $\beta_{\text{rot}} = 10^{-2}$ for rotations and $\beta_{\text{xyz}} = 10^0$ for translations across all experiments, which is the same optimizer setup we use for estimating camera poses in Eq. (17). We use the same gmNCC metric to compute image similarity in the objective function (7).

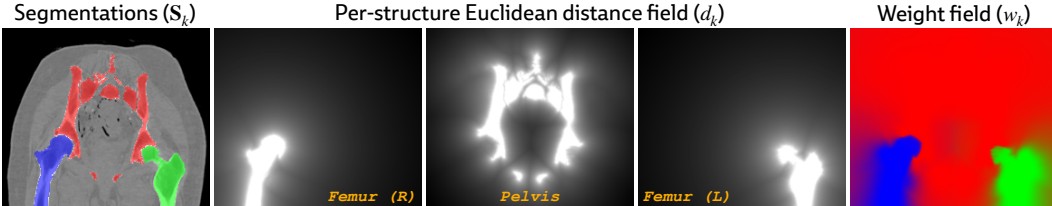

Figure 8: **A slice of the weight field produced by Eq. (6).** We visualize the weight field as the relative contribution of each structure at every pixel in the slice.

# D Implementations of Baselines

Below, we detail the implementation of all baselines compared to in this work. Note that all baselines, except for xvr, depend on accurate camera pose estimates, but do not specify protocols for calibrating the input X-ray images. Therefore, all methods (including our own) were evaluated using the same camera poses that we estimated using PolyPose.

**xvr** [16] is a rigid 2D/3D registration framework comprising (1) a patient-specific neural network pretrained on synthetic data to produce accurate initial pose estimates and (2) a test-time optimization protocol to refine initial pose estimates. We train patient-specific neural networks and perform test-time optimization for each subject using the default architecture and training hyperparameters.

**LiftReg** [18] is a deep dictionary learning method for deformable 2D/3D registration. It uses PCA to construct a low-rank vector space of 3D deformations given a dataset of patients with multiple CTs. Since patients in DeepFluoro do not have multiple CT scans, we can only evaluate LiftReg on the Head&Neck dataset. Specifically, we use FireANTs [37] to compute ground truth 3D deformations from pairs of CTs in the training set of Head&Neck. Then, we train a CNN to regress coefficients of the basis vectors, reconstruct the resulting deformation field, warp the moving CT, and compute the loss using 3D MSE and a diffusion regularizer.

**2D3D-RegNet** [19] uses a VoxelMorph-style [61] architecture to directly estimate a 3D deformation field given a moving CT and a fixed CBCT reconstructed from the input 2D X-rays. It is supervised using an image similarity loss on X-rays rendered from the warped CTs and the real X-rays, as well as an inverse consistency regularizer and an energy regularizer.

**uniGradICON** [34] and **multiGradICON** [35] are foundation models for intra- and inter-modality registration, respectively, trained on large datasets. We use the pretrained models available in their repositories in our experiments. These neural networks do not have any hyperparameters, and we optimize hyperparameters for their iterative variants on the validation set.

**FireANTs** [37] **and anatomix** [38] are improved solvers for 3D/3D registration that leverage novel optimization techniques and feature representations. We install the binaries available in their respective repositories and optimize the requisite hyperparameters on the validation set.

**Dense** $\mathbb{R}^3$ [23] places an optimizable displacement vector at every voxel in the moving CT scan. Similarly, Dense $\mathbf{SE}(3)$ [32] places an optimizable rototranslation generator at every voxel. We optimize both dense parameterizations with the same differentiable rendering setup as in PolyPose.

# E Additional Results

We present further visualizations of the Head&Neck and DeepFluoro datasets and a new analysis of a lung radiotherapy dataset with extreme soft tissue deformation.

## E.1 Head&Neck

In Figure 9, we render the deformation fields produced by 2D3D-RegNet [19] and multiGradICON (with and without IO) [35] for the same subject visualized in Figure 4. The warps produced by 2D3D-RegNet are well-behaved from a topological perspective, but fail to accurately capture the inter-scan motion of the patient's head. Deformations produced by the multiGradICON variants display an interesting failure mode, with the warp field radiating away from the isocenter of the CT scan. This dilation results in the anatomically implausible warped volumes visualized in Figure 4A.

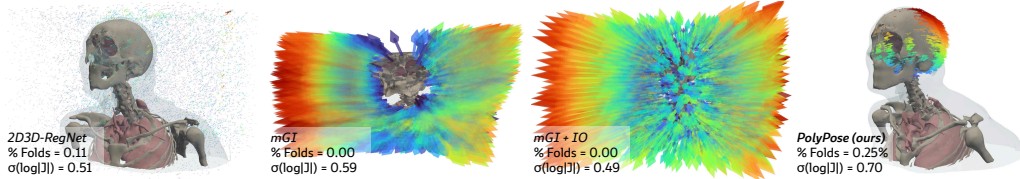

*2D3D-RegNet*
% Folds = 0.11
σ(log|J|) = 0.51

*mGI*
% Folds = 0.00
σ(log|J|) = 0.59

*mGI + IO*
% Folds = 0.00
σ(log|J|) = 0.49

*PolyPose (ours)*
% Folds = 0.25%
σ(log|J|) = 0.70

Figure 9: **3D renderings of the deformation fields produced by 2D3D-RegNet [19] and multi-GradICON [35].** These visualizations are complementary to the examples shown in Figure 4.

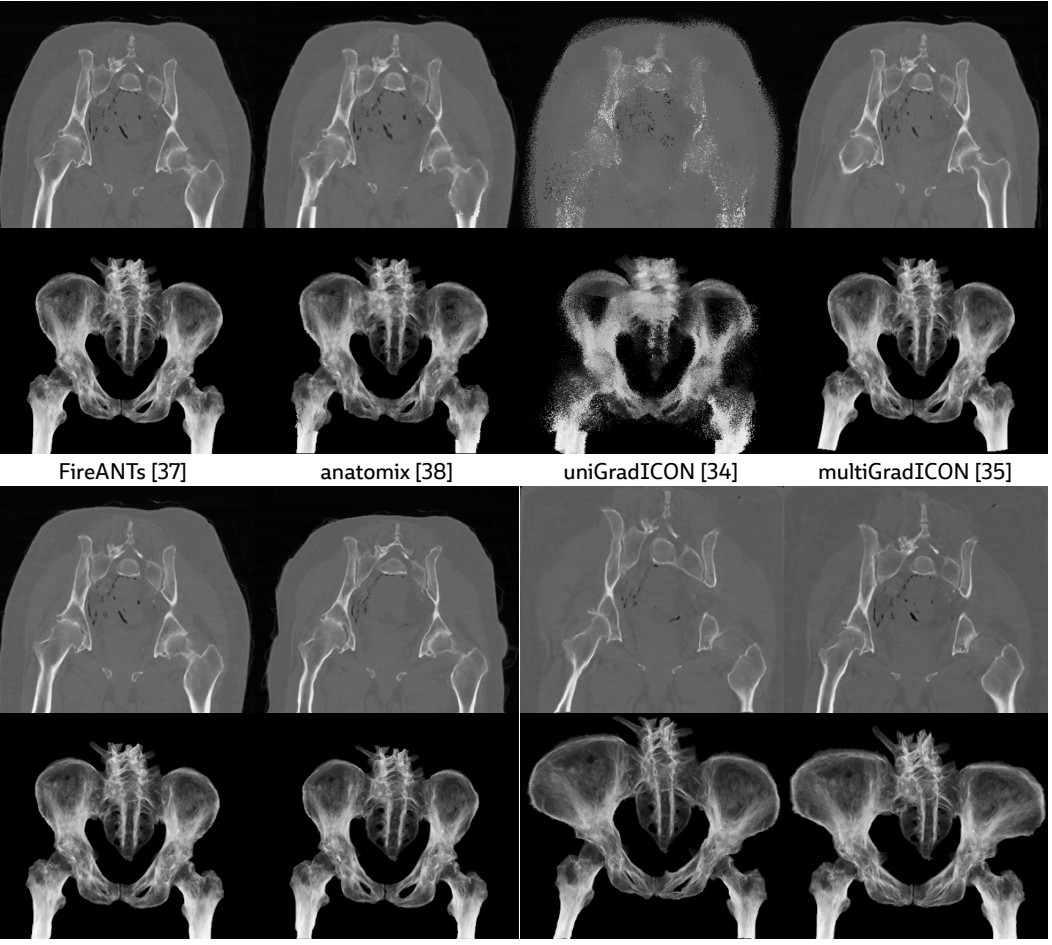

Figure 10: **Visualizations of the warped CTs produced by various methods on the same subject**. The *top rows* visualize central slices of the warped CTs and the *bottom rows* visualize maximum intensity projections along the coronal direction. Only PolyPose successfully recovers the anatomical motion (external rotation of the femurs) from minimal supervision (two X-ray images).

## E.2 DeepFluoro

In Figure 10, we visualize central slices and maximum intensity projections of the warped CTs produced by PolyPose as well as the baseline methods. This figure exemplifies many of the common failure modes for previous 2D/3D and 3D/3D registration methods:

- Dense parameterizations of the deformation field, such as $\mathbb{R}^3$ [23] and $\mathbf{SE}(3)$ [32, 42], can only influence voxels on which they have direct pixel supervision. As such, both of these methods break the femurs, outlining the bounded subregion of the preoperative volume that can be deformed.

- Both FireANTs [37] and anatomix [38] produce very small deformations, failing to capture the inter-scan motion of the patient. While the deformations these methods produce are not physiologically implausible, they are inaccurate and not useful in real-world settings.

- While uniGradICON [34] comes the closest of all the baselines to recovering the motion of the femurs, both it and multiGradICON [35] are affected by the streaking artifacts present in sparse-view CBCT reconstructions and produce dramatic dilations of the preoperative volume. This is analogous to the dilating warps produced by multiGradICON in the Head&Neck dataset (Figures 4 and 9).

## E.3 ThoraxCBCT

We perform additional experiments on a dataset of 13 patients undergoing chemoradiotherapy for advanced, non-small cell lung cancer [68]. Each patient had one pre-therapeutic fan-beam CT (FBCT) volume, used to plan the radiotherapy, and two intra-interventional cone-beam CT (CBCT) volumes, used to reorient the plan to the patient's current position and morphology [69]. Specifically, we registered pre-therapeutic FBCTs to synthetic X-rays generated from interventional CBCTs. This dataset is particularly challenging given the significant non-rigid respiratory motion between *maximum inhalation* (FBCT) and *maximum expiration* (CBCT) acquisitions.

Figure 11 visualizes exemplar registrations for the ThoraxCBCT dataset and reports quantitative evaluation metrics. Despite the substantial non-rigid deformation present in these volumes, PolyPose achieved superior registration accuracy compared to state-of-the-art 2D/3D and 3D/3D baselines. Notably, by modeling the motion of rigid bodies, PolyPose successfully captured the deformation of the soft tissue structures in the scene, including the liver, heart, aorta, trachea, esophagus, and all lobes of the lungs. This evaluation further demonstrates that the skeleton provide sufficient geometric constraints for accurate soft tissue registration, even under extreme respiratory conditions.

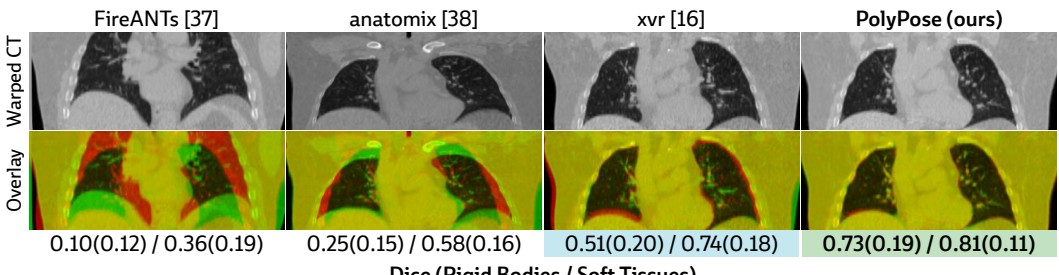

Figure 11: **PolyPose accurately models extreme deformations, including respiratory motion.** PolyPose achieves the highest 3D Dice for both rigid bodies and soft tissue, respectively.

## F Failure Cases

By construction, PolyPose produces diffeomorphisms. While this is intentional and generally a desirable property (the majority of human motion is smooth and invertible), diffeomorphisms do not represent all types of motion. We visualize this failure mode using a CT scan from an internal dataset of neurosurgical patients where the patient's mouth is closed in the preoperative CT, while it is open in the intraoperative X-rays. To represent opening the jaw, PolyPose repositions the patient's mandible in the warped CT. However, as the patient's top and bottom rows of teeth were touching in the preoperative CT, this downward warp creates an anatomically implausible stretching of the teeth in the lower jaw (see the red box in Figure 12). This is because, as diffeomorphisms are invertible, they cannot model the creation of empty space as occurs when the mouth opens. This defect results in the creation of a third row of teeth, as seen in the volume rendering.

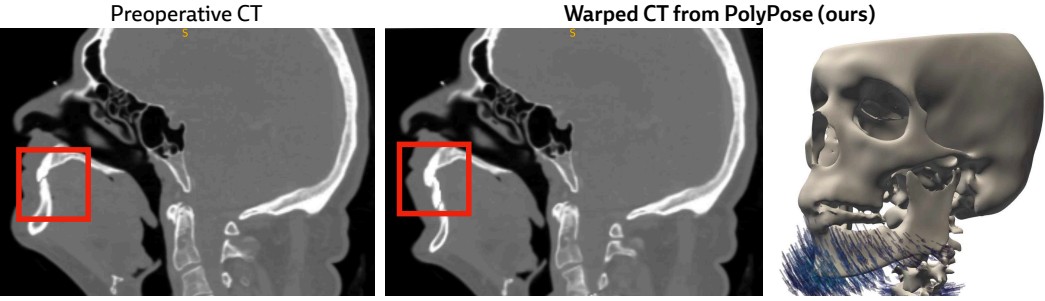

Figure 12: **An exemplar failure mode of diffeomorphisms.** The diffeomorphisms produced by PolyPose cannot represent certain motions, such as the opening of the mouth, as the top and bottom rows of teeth are touching in the preoperative CT scan and would require the creation of topologically-inconsistent empty space to match the target intraoperative X-rays.

