# OpenReview forum: "PolyPose: Deformable 2D/3D Registration via Polyrigid Transformations"
_NeurIPS.cc/2025/Conference — NeurIPS 2025 poster_

### Official Review · Reviewer_5y16 · 2025-07-02

**Clarity:** 3
**Significance:** 3
**Originality:** 3
**Rating:** 4
**Confidence:** 3

**Summary:**

This paper proposes PolyPose, a novel method for estimating the 3D pose and shape of deformable anatomical structures from sparse 2D X-ray projections. The authors introduce a polyrigid transform model, which decomposes the deformable object into rigid parts and allows independent pose estimation for each part. The method leverages a set of 2D observations and a learned canonical template to localize 3D points via pose-parameterized correspondences. To enable differentiable supervision, the framework integrates a custom X-ray rendering pipeline and a visibility network. The method is validated on human spine and pelvis data from clinical X-ray images and outperforms previous methods in both pose estimation and 3D reconstruction quality.

**Questions:**

1、Can the method generalize to anatomies without clear rigid substructures?
2、How does performance change with fewer 2D views?
3、Are the polyrigid parts interpretable in anatomical terms? 4. Could prior knowledge or statistical shape models help?
4、Could prior knowledge or statistical shape models help?
5、Anatomical plausibility of the weighting function: Is the assumption of uniform bone mass reasonable?

**Ethical Concerns:**

["NO or VERY MINOR ethics concerns only"]

**Final Justification:**

I think the quality of the article is quite good, which basically solves my doubts. I have decided to maintain my score.

**Limitations:**

The paper briefly acknowledges the potential limitation on generalization and requirement of predefined parts, but lacks a full discussion on modeling soft-tissue deformations, dependence on number of X-rays, or robustness to noise/occlusion.

**Quality:**

3

**Strengths And Weaknesses:**

Strengths:
- Innovative approach:Introducing multi-rigid body transformation into 2D/3D registration and Use anatomical priors to reduce the number of parameters for optimization degrees of freedom.
- Methodology: Differentiable X-ray renderer and visibility network are valuable contributions.
- Empirical results: Strong benchmarks and ablation studies.
- Clinical relevance: Supports low-dose imaging workflows.

Weaknesses:
- Scalability: Generalization to other anatomies (especially soft tissue) is untested.
- Manual part annotation required.
- Limited dataset and anatomical diversity.
- Real-time performance is not fully evaluated.

---

> ### Author Rebuttal · Authors · 2025-07-30
>
> We thank the reviewer for their feedback and thoughtful questions! Their feedback and the responses below will be integrated into the manuscript. We are glad they found our approach to this clinically relevant problem to be innovative and well-supported by strong experimental results.
>
> > _”Weakness: Generalization to other anatomies (especially soft tissue) is untested.”_
>
> This is a potential miscommunication. We specifically test generalization to soft tissue structures in our experiment on the Head&Neck cancer dataset (Figure 5). These experiments demonstrate that our method achieves state-of-the-art accuracy in the localization of soft tissues. We will clarify the text to emphasize this point further.
>
> > _”Weakness: Manual part annotation required.”_
>
>
> We respectfully clarify that manual annotation is not needed for our work. PolyPose only needs bone annotations from CT volumes, which are easily obtained using off-the-shelf tools such as TotalSegmentator [64]. Segmentation can thus be performed automatically, requiring only a few seconds per volume without any manual intervention. For example, our experiments on the Head&Neck cancer dataset only use TotalSegmentator for annotating rigid bodies.
>
> > _”Weakness: Limited dataset and anatomical diversity.”_
>
>
> We respectfully posit that our evaluation datasets come from highly disparate regions of the body with no overlap, such as the pelvis and the head-neck-shoulder complex in the DeepFluoro and Head&Neck cancer datasets, respectively.
>
> > _”Weakness: Real-time performance is not fully evaluated.”_
>
> Thank you for raising this. We report the runtimes of all methods evaluated in our paper on the Head&Neck cancer dataset in the table below and will include this information in the paper. While PolyPose is slower than the inference-only LiftReg and 2D3D-RegNet methods, it requires no training and is much more accurate. Runtimes and Dice metrics are averaged over all numbers of views (2-9) evaluated in Figure 5. All methods are evaluated on a single NVIDIA RTX A6000 GPU.
>
> |                  | **Dice (Rigid Bodies)** | **Dice (Soft Tissues)** | **Runtime (min:sec)** |
> |------------------|:-----------------------:|:-----------------------:|:---------------------:|
> | PolyPose         |        0.71(0.15)       |        0.76(0.08)       |       0:59(0:20)      |
> | FireANTS         |        0.39(0.13)       |        0.52(0.17)       |       0:41(0:08)      |
> | anatomix         |        0.57(0.14)       |        0.66(0.13)       |       1:05(0:12)      |
> | multiGradICON    |        0.03(0.02)       |        0.09(0.03)       |       0:28(0:03)      |
> | multiGradICON+IO |        0.20(0.14)       |        0.32(0.17)       |       1:10(0:08)      |
> | 2D3D-RegNet      |        0.58(0.12)       |        0.67(0.09)       |       0:01(0:01)      |
> | LiftReg          |        0.47(0.13)       |        0.57(0.12)       |       0:02(0:01)      |
> | DiffPose         |        0.53(0.15)       |        0.61(0.13)       |       0:06(0:02)      |
>
> > _”Question: Can the method generalize to anatomies without clear rigid substructures?”_
>
> This is an important question. In our experiments on the Head&Neck dataset, our method accurately generalizes to and aligns soft tissue structures without explicitly modeling them. We do not claim that our solution applies to the extreme scenario where there are no skeletal structures available to constrain the deformation field. We will further emphasize this in the discussion.
>
> > _”Question: How does performance change with fewer 2D views?”_
>
> We conducted this experiment in Figure 5, which shows that PolyPose’s performance remains robust as the number of 2D views/X-rays decreases. Compared to the baselines, our method has the highest accuracy across all numbers of views for both rigid and soft tissue structures.
>
> > _”Question: Are the polyrigid parts interpretable in anatomical terms?”_
>
> This is an important advantage of our method that we did not emphasize in the initial submission. Thank you for raising it! Yes, the polyrigid parts in our work are the bones in the patient’s skeleton, which leads to the estimated deformation fields being interpretable in anatomical terms by construction. For example, in Figure 2 (right), each displacement vector is colored by the relative contribution of the rigid movement of each bone. One can observe that the estimated warp field is *locally-linear*, with the deformation vectors being most influenced by the nearest rigid structures (Figure 2).
>
> > _”Question: Could prior knowledge or statistical shape models help?”_
>
> We agree that prior knowledge helps our problem context. To re-emphasize, our model already incorporates prior knowledge through the biological constraint that individual bones do not bend in typical motion. As a result, our polyrigid deformation model automatically enforces anatomically plausible priors that respect the piecewise rigid nature of human movement. This is why our method does not require any explicit regularization in such an ill-posed and underdetermined setting. Statistical shape models could also be potentially integrated in future work, but we found volumetric bone segmentations to be easy to obtain and to provide robust results in our framework. We would be happy to discuss any alternative priors or parameterizations the reviewer is interested in during the discussion period.
>
> > _”Question: Anatomical plausibility of the weighting function: Is the assumption of uniform bone mass reasonable?”_
>
> Within the context of registration, we believe that assuming uniform density within an individual bone is a reasonable approximation. While mineral density can change subtly across subregions of a bone, this variation is not well characterized in the literature and would require specialized imaging, such as a DEXA scan for each subject. As demonstrated by the experiments, our simplifying assumption is robust across multiple anatomical regions and leads to large improvements over existing baselines.

---

> > ### Comment · Reviewer_5y16 · 2025-08-05
> > **Comments on PolyPose Rebuttal**
> >
> > Your rebuttal addresses most of the key concerns raised in the review, particularly regarding runtime, annotation efficiency, and anatomical interpretability. I still recommend that the manuscript include more explicit discussion on generalization beyond rigid structures and robustness to soft-tissue-heavy or occluded input cases, especially in anatomies where skeletal anchors are limited or absent, which may challenge the method’s assumptions. That said, I have decided to maintain my original score, as the core concerns around generalization and dataset diversity remain only partially addressed.

---

> > > ### Author Response · Authors · 2025-08-06
> > >
> > > We are glad to hear that we have addressed the majority of the reviewer’s key concerns\! We will clarify in the manuscript that our method is designed for anatomical regions containing skeletal structures and does not claim applicability to purely soft-tissue anatomies.
> > >
> > > To address the reviewer's remaining concern regarding generalization to scenarios with significantly more soft tissue, we conducted additional evaluations on the ***ThoraxCBCT*** dataset, which contains thoracic fan-beam CT (FBCT) and cone-beam CT (CBCT) volumes for radiotherapy planning. Specifically, we registered pre-therapeutic FBCTs to synthetic X-rays generated from interventional CBCTs. This dataset is particularly challenging given the extreme non-rigid respiratory motion between *maximum inhalation* (FBCT) and *maximum expiration* (CBCT) acquisitions.
> > >
> > > Despite this substantial non-rigid deformation, our method achieved superior registration accuracy compared to state-of-the-art 3D/3D baselines. Notably, by modeling the motion of rigid bodies, our approach successfully captured the deformation of the soft tissue structures in the scene, including the lungs, heart, aorta, trachea, and esophagus. This evaluation provides additional evidence that skeletal structures provide sufficient geometric constraints for accurate soft tissue registration, even under challenging respiratory conditions.
> > >
> > > |    Method    | Dice - Rigid Bodies | Dice - Soft Tissues |
> > > |:------------:|:-------------------:|:-------------------:|
> > > | **PolyPose** |   **0.731(0.191)**  |   **0.806(0.112)**  |
> > > |   anatomix   |     0.247(0.147)    |     0.581(0.159)    |
> > > |   FireANTs   |     0.096(0.117)    |     0.364(0.189)    |
> > >
> > > - **Rigid Bodies:** sternum, scapula (L/R), clavicles (L/R), T-spine, ribs (L/R)
> > > - **Soft Tissues:** all lung lobes, heart, aorta, liver, trachea, and esophagus
> > >
> > > We will incorporate our evaluation on this additional diverse dataset into the revised manuscript. We are grateful for the reviewer’s time and feedback, and welcome any further discussion\!

---

> > > > ### Comment · Reviewer_5y16 · 2025-08-09
> > > >
> > > > Thanks to the author for his hard work, I think my concerns have been resolved

---

### Official Review · Reviewer_Jfu5 · 2025-07-02

**Clarity:** 3
**Significance:** 2
**Originality:** 3
**Rating:** 4
**Confidence:** 3

**Summary:**

This paper presented PolyPose for deformable 2D/3D registration. PolyPose parameterizes complex 3D deformation fields as a composition of rigid transforms, leveraging the biological constraint that individual bones do not bend in typical motion. The model enforces anatomically plausible priors that respect the piecewise rigid nature of human movement. PolyPose solves sparse-view deformable registration and aligns preoperative 3D images to intraoperative 2D X-rays using a locally rigid deformation model. It requires only X-rays and eliminates complex regularization. Extensive experiments on diverse datasets from orthopedic surgery and radiotherapy align the patient’s preoperative volumes to two X-rays.

**Questions:**

1. The model relied on the assumption that anatomical structures are rigid, being limited to bones with subtle bending or complex joint interactions. Whether the proposed model can be extended to soft tissue deformation and non-skeletal contexts?
2. The model required precise segmentation of rigid bodies in the preoperative volumetric images. The segmentation errors can cause inaccurate deformation fields. The manual and semi-automatic segmentation impose additional time complexity.
3. The initial camera poses relied on rigid registration of an anchor structure. The obscured anchor structure can cause pose estimation errors.
4. Tangent space averaging (Eq. 4) assumes transforms combine linearly in se(3). It is unclear whether such computation can handle large-scale transformations or deformations.

**Ethical Concerns:**

["NO or VERY MINOR ethics concerns only"]

**Final Justification:**

The authors have provided detailed clarifications to main concerns in the review, including those regarding the rigid anatomical structure alignment, segmentation of rigid bodies in CT/X-rays, and camera pose estimation. Additional discussions on the model performances under different strengths of corruption and extrapolation to soft tissue structures help improve clarity. I still suggest that this work has discussions on structures that are not strictly rigid. Discussions on the reliability of extrapolating to soft tissue structures would be helpful in demonstrating the generalization capacity to diverse structures. Considering the overlapping of anatomical structures on 2D X-rays and image artifacts, the discussions on the effects of segmentation errors of anchor structures and rigid bodies are required to validate the proposed methods.

**Limitations:**

yes

**Quality:**

2

**Strengths And Weaknesses:**

Strengths
This paper presented a regularization-free framework for deformable 2D/3D registration that estimates polyrigid deformation fields using differentiable X-ray rendering.　The proposed anatomically motivated motion model is robust in sparse-view and limited-angle settings and produces smooth, invertible, and accurate deformation fields through construction.

Weaknesses
The model relied on the assumption that anatomical structures are rigid. It would be helpful to discuss its generalization capacity when confronted with subtle bending or complex joint interactions. Moreover, the model relied on manual or semi-supervised segmentation of rigid bodies in the preoperative volumetric images and inaccurate deformation fields. The initial camera pose estimation relied on rigid registration of an anchor structure.

---

> ### Author Rebuttal · Authors · 2025-07-30
>
> Thank you for the feedback and detailed questions and comments – they are addressed individually below. The question about robustness to segmentation errors led to an interesting ablation study that we will include in the updated manuscript. We are also glad the reviewer appreciated the anatomical motivation behind our method, which leads to a regularization-free framework with strong priors.
>
> > _”The model relied on the assumption that anatomical structures are rigid, being limited to bones with subtle bending or complex joint interactions. Can the proposed model can be extended to soft tissue deformation and non-skeletal contexts?”_
>
> This may be a potential miscommunication. Our method does not explicitly represent soft tissue deformations. However, soft tissue structures directly contribute to the image-based loss used during registration (Figure 3B) and are thus incorporated into the deformation estimation. To clarify, our only assumption is that bones move rigidly; no rigidity is assumed for other structures.
>
> In practice, methods that do not have this prior produce unrealistic and suboptimal results in the highly ill-posed setting of CT to X-ray registration (Figure 4), as soft tissue structures are very difficult to distinguish in X-ray. In contrast, our anatomically motivated prior on human motion enables PolyPose to successfully extrapolate to soft tissue structures that were not explicitly modeled and achieve the best overall alignment (Figure 5).
>
> We do not claim applicability to CT to X-ray registration tasks that do not have identifiable rigid substructures. We will clarify this in the revised discussion, thank you for the clarifying question.
>
> > _”The model required precise segmentation of rigid bodies in the preoperative volumetric images. The segmentation errors can cause inaccurate deformation fields.”_
>
> This is an important question to investigate – thank you for raising it. To simulate errors typical of automated segmentation tools like TotalSegmentator, we present an experiment below where we systematically erode the ground truth segmentations in the DeepFluoro experiments by increasing the radius of erosion. Here, radius=0mm corresponds to no erosion and is equivalent to the experiment in Table 1 (registration given two X-rays from a limited viewing angle). We find that as we increase the strength of corruption, our method remains robust against extreme corruption and still outperforms the baselines in Table 1 both in terms of Dice and Hausdorff Distance.
>
> | **Erosion Radius** | **Pelvis - Dice (↑)** | **Pelvis - HD95 (↓)** | **Femur (L) - Dice (↑)** | **Femur (L) - HD95 (↓)** | **Femur (R) - Dice (↑)** | **Femur (R) - HD95 (↓)** |
> |--------------------|-----------------------|-----------------------|--------------------------|--------------------------|--------------------------|--------------------------|
> | 0 mm               |       0.99(0.00)      |       1.00(0.00)      |        0.99(0.00)        |        1.02(0.10)        |        0.98(0.00)        |        1.43(0.36)        |
> | 1 mm               |       0.99(0.00)      |       1.00(0.00)      |        0.98(0.00)        |        1.11(0.49)        |        0.98(0.00)        |        1.43(0.46)        |
> | 2 mm               |       0.99(0.00)      |       1.00(0.00)      |        0.98(0.00)        |        1.07(0.24)        |        0.98(0.01)        |        1.66(0.43)        |
> | 3 mm               |       0.99(0.00)      |       1.00(0.00)      |        0.98(0.00)        |        1.15(0.27)        |        0.97(0.01)        |        1.55(0.45)        |
> | 4 mm               |       0.99(0.00)      |       1.01(0.07)      |        0.97(0.00)        |        1.71(0.50)        |        0.97(0.01)        |        1.84(0.42)        |
> | 5 mm               |       0.99(0.00)      |       1.11(0.23)      |        0.95(0.00)        |        3.20(1.25)        |        0.96(0.01)        |        2.28(0.69)        |
> | 6 mm               |       0.99(0.00)      |       1.21(0.34)      |        0.95(0.01)        |        2.89(1.18)        |        0.96(0.01)        |        2.95(1.76)        |
> | 7 mm               |       0.99(0.00)      |       1.41(0.46)      |        0.94(0.01)        |        3.64(1.27)        |        0.95(0.01)        |        3.89(3.08)        |
> | 8 mm               |       0.99(0.00)      |       1.64(0.49)      |        0.93(0.01)        |        5.13(3.19)        |        0.95(0.01)        |        3.70(3.17)        |
> | 9 mm               |       0.99(0.00)      |       1.83(0.61)      |        0.94(0.01)        |        3.47(1.50)        |        0.94(0.01)        |        4.24(3.36)        |
>
> The results are stable to 4 mm of erosion, beyond which point the accuracy degrades gracefully. However, an erosion radius of just 4mm corresponds to a significant corruption of the original segmentation with its volume decreasing by 40%-60%. For example, the volume of the left femur label decreases from 186,953 mm³ to 117,440 mm³ with an erosion radius of 4 mm.
>
> > _” The manual and semi-automatic segmentation impose additional time complexity.”_
>
> We respectfully clarify that manual annotation is not needed for our work and for the majority of use cases in radiotherapy and orthopedic surgery. PolyPose only uses bone annotations on CT volumes, which can easily be segmented using off-the-shelf tools such as TotalSegmentator [64]. This automatic process requires only a few seconds per volume without any manual intervention, adding minimal time complexity.
>
> > _”The initial camera poses relied on rigid registration of an anchor structure. The obscured anchor structure can cause pose estimation errors.”_
>
> We agree that the choice of anchor structure is critical to ensure the success of the initial rigid registration used for preprocessing. In CT and X-ray images, most bones are clearly visible and not obscured in practice due to their higher X-ray attenuation and we only use bones as rigid structures in our formulation. For example, this is why we selected the pelvis for our DeepFluoro experiments, as it is both central and clearly identifiable. We will update the manuscript to include this recommendation for readers.
>
> > _”Tangent space averaging (Eq. 4) assumes transforms combine linearly in se(3). It is unclear whether such computation can handle large-scale transformations or deformations.”_
>
> We respectfully clarify that, as the tangent space is a Euclidean vector space (and not an embedded manifold like SE(3)), it is valid to linearly combine rigid transforms represented in se(3). As a result, Equation 4 produces diffeomorphisms by construction [26, 51], meaning our method can model everything from a global rigid transform to a large-scale dense deformation field.

---

> > ### Comment · Reviewer_Jfu5 · 2025-08-05
> > **Comments on PolyPose Rebuttal**
> >
> > Thanks for the detailed clarifications to main concerns in the review, including those regarding the rigid anatomical structure alignment, segmentation of rigid bodies in CT/X-rays, and camera pose estimation. Additional discussions on the model performances under different strengths of corruption and extrapolation to soft tissue structures help improve clarity. I still suggest that this work has discussions on structures that are not strictly rigid. Discussions on the reliability of extrapolating to soft tissue structures would be helpful in demonstrating the generalization capacity to diverse structures. Considering the overlapping of anatomical structures on 2D X-rays and image artifacts, the discussions on the effects of segmentation errors of anchor structures and rigid bodies are required to validate the proposed methods. I have no further questions at this time.

---

> > > ### Author Response · Authors · 2025-08-06
> > >
> > > We thank the reviewer for their follow up and are glad to hear that we have addressed the main concerns of their review. We will add discussions regarding the robustness of our method to segmentation errors from our rebuttal to the final manuscript. We will also explicitly clarify in the final manuscript that our method does not apply to scenarios where no skeletal structures are present.
> > >
> > > To address the reviewer's remaining concern regarding generalization to diverse soft tissue structures, we conducted additional evaluations on the ***ThoraxCBCT*** dataset, which contains thoracic fan-beam CT (FBCT) and cone-beam CT (CBCT) volumes for radiotherapy planning. Specifically, we registered pre-therapeutic FBCTs to synthetic X-rays generated from interventional CBCTs. This dataset is particularly challenging given the extreme non-rigid respiratory motion between *maximum inhalation* (FBCT) and *maximum expiration* (CBCT) acquisitions.
> > >
> > > Despite this substantial non-rigid deformation, our method achieved superior registration accuracy compared to state-of-the-art 3D/3D baselines. Notably, by modeling the motion of rigid bodies, our approach successfully captured the deformation of the soft tissue structures in the scene, including the lungs, heart, aorta, trachea, and esophagus. This evaluation provides additional evidence that skeletal structures provide sufficient geometric constraints for accurate soft tissue registration, even under challenging respiratory conditions.
> > >
> > > |    Method    | Dice - Rigid Bodies | Dice - Soft Tissues |
> > > |:------------:|:-------------------:|:-------------------:|
> > > | **PolyPose** |   **0.731(0.191)**  |   **0.806(0.112)**  |
> > > |   anatomix   |     0.247(0.147)    |     0.581(0.159)    |
> > > |   FireANTs   |     0.096(0.117)    |     0.364(0.189)    |
> > >
> > > - **Rigid Bodies:** sternum, scapula (L/R), clavicles (L/R), T-spine, ribs (L/R)
> > > - **Soft Tissues:** all lung lobes, heart, aorta, liver, trachea, and esophagus
> > >
> > > We will incorporate our evaluation on this additional diverse dataset into the revised manuscript and are grateful for the reviewer’s time and feedback.

---

### Official Review · Reviewer_CMp5 · 2025-07-03

**Clarity:** 3
**Significance:** 3
**Originality:** 3
**Rating:** 4
**Confidence:** 4

**Summary:**

In this paper, the authors tried to address the problem of deformable 2D/3D registration between a preoperative 3D scan (e.g. a volumetric CT) with 2D X-ray images acquired during the procedure (intra-operative). Instead of going full rigid or full deformable, they new idea is to leverage an anatomical prior that individual bones move as rigid bodies to simplify the deformation model. By combining these local rigid motions, the method produces a locally rigid warp field that is smooth and diffeomorphic by construction. A differentiable X-ray rendering procedure is used o compare the warped 3D volume against the 2D X-rays, allowing gradient-based optimization of the rigid body poses. The authors demonstrates that the proposed approach, PolyPose, can align a patient's preoperative 3D volume to as few as two 2D X-rays, even from limited angles, without additional regularization. Extensive experiments on diverse datasets from orthopedic surgery and radiotherapy show that PolyPose outperforms prior approaches.

**Questions:**

- The method relies on treating certain anatomical parts as rigid bodies. How are these rigid structures defined and obtained in practice? (Manually?)
- As mentioned, it would be helpful to analyze the robustness of the proposed method against segmentation errors.
- In the limited-angle scenarios, some bones might be barely visible or even completely occluded in the available X-ray views. How does the model handle a rigid component with little direct 2D supervision? And since there is no explicit regularization, it would be helpful to discuss this corner cases
- The proposed method didn't optimize soft tissue motion, and such motion are carried along by blended warp field. The results show improved alignment of some soft organs without modeling them. However, one might wonder about scenarios with significant non-rigid soft tissues change? Is this still under scope of PolyPose?

**Ethical Concerns:**

["NO or VERY MINOR ethics concerns only"]

**Limitations:**

yes

**Paper Formatting Concerns:**

No concern

**Quality:**

4

**Strengths And Weaknesses:**

Strengths
- A novel approach to apply a locally-rigid deformation model in the 2D/3D registration setting.
- The model can work as few as 2 X-ray images, and with limited angle scenarios
- The proposed method does not require expensive deformation regularization or hyperparameter tuning. This greatly improves the ease of use in real world applications.
- Strong performance comparing to deep learning based or optimization-based methods.
- The code is publicly available with all implementation details.

Weakness
- The proposed method requires accurate segmentation of rigid bones. A study of how the performance depends on segmentation accuracy would be helpful. While there are many segmentation tools nowadays, they still make notable errors from time to time in real world application, esp. on diseased situation or anomality.
- As the authors mentioned, diffeomorphic deformation cannot be handled here, such as jaw opening. In fact, during neurosurgery, there might be minor deformation of the patient's head against the pre-operative scan, due to the head clamp. In other words, it is unclear how polypose can handle such situation when there exists even minor non-rigid deformation in the local rigid piece of bones.
- The paper does not explicitly discuss runtime or computational efficiency, which could be important for intraoperative use.

---

> ### Author Rebuttal · Authors · 2025-07-30
>
> We thank the reviewer for their insightful questions. These led to interesting additional experiments that have improved our understanding of the robustness of PolyPose. We are also glad that they found our novel approach to be extensively validated and easy to use in real-world applications due to its lack of expensive regularization or optimizable hyperparameters.
>
> > _”The proposed method requires accurate segmentation of rigid bones. A study of how the performance depends on segmentation accuracy would be helpful. While there are many segmentation tools nowadays, they still make notable errors from time to time in real world application, esp. on diseased situation or anomality.”_
>
> This is an important question to investigate – thank you for raising it. To simulate errors typical of automated segmentation tools like TotalSegmentator, we present an experiment below where we systematically erode the ground truth segmentations in the DeepFluoro experiments by increasing the radius of erosion. Here, radius=0mm corresponds to no erosion and is equivalent to the experiment in Table 1 (registration given two X-rays from a limited viewing angle). We find that as we increase the strength of corruption, our method remains robust against extreme corruption and still outperforms the baselines in Table 1 both in terms of Dice and Hausdorff Distance.
>
> | **Erosion Radius** | **Pelvis - Dice (↑)** | **Pelvis - HD95 (↓)** | **Femur (L) - Dice (↑)** | **Femur (L) - HD95 (↓)** | **Femur (R) - Dice (↑)** | **Femur (R) - HD95 (↓)** |
> |--------------------|-----------------------|-----------------------|--------------------------|--------------------------|--------------------------|--------------------------|
> | 0 mm               |       0.99(0.00)      |       1.00(0.00)      |        0.99(0.00)        |        1.02(0.10)        |        0.98(0.00)        |        1.43(0.36)        |
> | 1 mm               |       0.99(0.00)      |       1.00(0.00)      |        0.98(0.00)        |        1.11(0.49)        |        0.98(0.00)        |        1.43(0.46)        |
> | 2 mm               |       0.99(0.00)      |       1.00(0.00)      |        0.98(0.00)        |        1.07(0.24)        |        0.98(0.01)        |        1.66(0.43)        |
> | 3 mm               |       0.99(0.00)      |       1.00(0.00)      |        0.98(0.00)        |        1.15(0.27)        |        0.97(0.01)        |        1.55(0.45)        |
> | 4 mm               |       0.99(0.00)      |       1.01(0.07)      |        0.97(0.00)        |        1.71(0.50)        |        0.97(0.01)        |        1.84(0.42)        |
> | 5 mm               |       0.99(0.00)      |       1.11(0.23)      |        0.95(0.00)        |        3.20(1.25)        |        0.96(0.01)        |        2.28(0.69)        |
> | 6 mm               |       0.99(0.00)      |       1.21(0.34)      |        0.95(0.01)        |        2.89(1.18)        |        0.96(0.01)        |        2.95(1.76)        |
> | 7 mm               |       0.99(0.00)      |       1.41(0.46)      |        0.94(0.01)        |        3.64(1.27)        |        0.95(0.01)        |        3.89(3.08)        |
> | 8 mm               |       0.99(0.00)      |       1.64(0.49)      |        0.93(0.01)        |        5.13(3.19)        |        0.95(0.01)        |        3.70(3.17)        |
> | 9 mm               |       0.99(0.00)      |       1.83(0.61)      |        0.94(0.01)        |        3.47(1.50)        |        0.94(0.01)        |        4.24(3.36)        |
>
> The results are stable to 4 mm of erosion, beyond which point the accuracy degrades gracefully. However, an erosion radius of just 4mm corresponds to a significant corruption of the original segmentation with its volume decreasing by 40%-60%. For example, the volume of the left femur label decreases from 186,953 mm³ to 117,440 mm³ with an erosion radius of 4 mm.
>
> > _”“The method relies on treating certain anatomical parts as rigid bodies. How are these rigid structures defined and obtained in practice? (Manually?)”_
>
> Regarding the definition of rigid pieces, our only assumption is that bones do not bend in typical motion. Thus, bone segmentations are defined as rigid pieces, and no other anatomical regions are treated as rigid bodies.
>
> Regarding obtaining these bone (rigid body) segmentations on our CT volumes, individual bone segmentations are easily obtained using off-the-shelf tools such as TotalSegmentator [64]. This requires only a few seconds per volume without any manual intervention, adding minimal time complexity. For example, this is the exact procedure we used to obtain bone segmentations for PolyPose for the Head&Neck dataset.
>
> > _”As the authors mentioned, diffeomorphic deformation cannot be handled here, such as jaw opening. In fact, during neurosurgery, there might be minor deformation of the patient's head against the pre-operative scan, due to the head clamp. In other words, it is unclear how polypose can handle such situation when there exists even minor non-rigid deformation in the local rigid piece of bones.”_
>
> We respectfully clarify that our method *specifically* produces diffeomorphic deformations by construction. While each bone is assumed to move rigidly, the overall deformation field is estimated by a tangent space interpolation of rigid transforms that best fits the image alignment losses and does not require each bone to move with exact rigidity in the final deformation. Further, while diffeomorphisms cannot cover edge cases that are non-diffeomorphic (e.g., the jaw opening, a limitation that the reviewer noted from Appendix G), it is sufficiently accurate to describe subtle topological changes, like the indentations caused by a head clamp.
>
> Due to this diffeomorphic constraint, we do not claim to be accurate if there are major topological changes between the preoperative and intraoperative imaging, e.g., if the patient’s skull appears intact in the CT, but is fractured in the X-ray. These scenarios could be addressed in PolyPose by masking the loss function in regions without topological correspondence; however, this is beyond the scope of our work and  there is no publicly available paired CT/X-ray dataset with major topological changes for us to evaluate such a variant. We will expand the discussion section to emphasize these points—thank you for raising them.
>
> > _”The paper does not explicitly discuss runtime or computational efficiency, which could be important for intraoperative use.”_
>
> Thank you for raising this. We report the runtimes of all methods evaluated in our paper on the Head&Neck cancer dataset in the table below and will include this information in the paper. While PolyPose is slower than the inference-only LiftReg and 2D3D-RegNet methods, it requires no training and is much more accurate. Runtimes and Dice metrics are averaged over all numbers of views (2-9) evaluated in Figure 5. All methods are evaluated on a single NVIDIA RTX A6000 GPU.
>
> |                  | **Dice (Rigid Bodies)** | **Dice (Soft Tissues)** | **Runtime (min:sec)** |
> |------------------|:-----------------------:|:-----------------------:|:---------------------:|
> | PolyPose         |        0.71(0.15)       |        0.76(0.08)       |       0:59(0:20)      |
> | FireANTS         |        0.39(0.13)       |        0.52(0.17)       |       0:41(0:08)      |
> | anatomix         |        0.57(0.14)       |        0.66(0.13)       |       1:05(0:12)      |
> | multiGradICON    |        0.03(0.02)       |        0.09(0.03)       |       0:28(0:03)      |
> | multiGradICON+IO |        0.20(0.14)       |        0.32(0.17)       |       1:10(0:08)      |
> | 2D3D-RegNet      |        0.58(0.12)       |        0.67(0.09)       |       0:01(0:01)      |
> | LiftReg          |        0.47(0.13)       |        0.57(0.12)       |       0:02(0:01)      |
> | DiffPose         |        0.53(0.15)       |        0.61(0.13)       |       0:06(0:02)      |
>
> > _”In the limited-angle scenarios, some bones might be barely visible or even completely occluded in the available X-ray views. How does the model handle a rigid component with little direct 2D supervision? And since there is no explicit regularization, it would be helpful to discuss this corner cases”_
>
> This scenario occurs within our experiments on the DeepFluoro dataset, which is both limited-angle and has barely visible structures (the L/R femurs) due to the restricted field-of-view. PolyPose extrapolates well to these structures and outperforms existing widely-used methods for this task. For more details, please see Figure 6 and Table 1, as well as Figure 10 in Appendix E.2.
>
> > _”The proposed method didn't optimize soft tissue motion, and such motion are carried along by blended warp field. The results show improved alignment of some soft organs without modeling them. However, one might wonder about scenarios with significant non-rigid soft tissues change? Is this still under scope of PolyPose?”_
>
> Thank you for this interesting question! In general, PolyPose is effective for soft tissues that deform along with the movement of rigid structures (e.g., the esophagus and skull / cervical spine, the lungs’ upper lobes and the ribs / sternum / spine, etc.). For these cases, extreme soft tissue motion will also be coupled with significant motion of the rigid structures, which our experiments show can be faithfully modeled by PolyPose. We do not claim applicability to procedures that are entirely non-rigid with no rigid substructures visible that could be used as anchors. We will make sure to expand the limitations to discuss this.

---

### Official Review · Reviewer_Y5jy · 2025-07-03

**Clarity:** 4
**Significance:** 3
**Originality:** 3
**Rating:** 5
**Confidence:** 4

**Summary:**

The authors present PolyPose, a simple and robust method for deformable 2D/3D registration. PolyPose parameterizes complex 3D deformation fields as a composition of rigid transforms. First by estimating the camera matrcis through an anchored sub-structure that is reliably visible across all views and rigidly aligned, the authors get the accurate extrinsic matrix estimation for every angled X-ray image. Then, after obtaining the correct camera matrics, the authors construct the polyrigid deformation field by leveraging comparable sized structures as separate warped coordinate systems, and weighted by respective mass and distance controlled weight field for each structure and 2D X-ray point combination to represent the global rigid transformation.

**Questions:**

Please see my weakness section.

**Ethical Concerns:**

["NO or VERY MINOR ethics concerns only"]

**Final Justification:**

After reading authors' rebuttal and discussion with other reviewers, my concerns have been addressed and thanks for that.
However, this work is still limited to fixed setting of targeted 2d-3d scenarios, which hinders its general application to all tasks. Considering all above aspects, I decide to keep my original positive rating.

**Limitations:**

yes

**Quality:**

3

**Strengths And Weaknesses:**

Strength:
1. The novelty is clear and prominent. By simplifying a non-rigid patient and procedure specific transform into the polyrigid warps at all voxels of the 3D pre-volume which is decided by a linear mass and distance weighted combination of all rigid articulated bodies.
2. Estimating the camera matrices through an anchor structure is very clever. The criteria for selecting such anchor structure is also important, such as (a) is reliably visible across all views in X-ray 2D sparse registration, as well as (b) could be rigidly aligned via its static property, like pelvis in Figure 3A.
3. The formalization is clear and complete, although the methodology is complex, the description is clear and precise.
4. The experiment resutls clearly show the superiority of proposed technique, including larger overlay in dice score and lower deformation error and topological error in tangent space, for most views of X-ray sparse registration, over multiple baselines on two large body data sets.

Weakness:
1. In equation (6), why only structure mass and distance are taken into consideration, how about their size and the distance to the mass center point?
2. With equation (3) and rigidly aligned S_{anchor} which is available at all views, all angled camera matrices could be estimated, but how many sampled 2D points $p$ in X-ray images are needed for a correct evaluation for the X-ray projection operator P? And what is the confidence we should keep for such a discrete approximation of first-order approximation following Beer-Lambert law? Any alternative evaluation loop between camera matrics and acquisition deformation field needed?
3. Equation (9) I can understand the SE(3) to se(3) transformation, but why it is isomorphic to R^6?
4. For experiment, I am more interested in a general body data set where different articulated bodies or soft tissues are in different size and mass, and no rigid transform or global alignment guarantee for any body or structure.

---

> ### Author Rebuttal · Authors · 2025-07-30
>
> We thank the reviewer for their interesting mathematical questions, and are happy that they found our method to be clever and our formalization of a mathematically complex methodology to be clear, complete, and precise.
>
> > _”In equation (6), why only structure mass and distance are taken into consideration, how about their size and the distance to the mass center point?”_
>
> Regarding size, we want to respectfully clarify that our weight function (Equation 6) does take the relative size of rigid structures into consideration. For all rigid bodies, we assume that bone density is constant (Line 156), such that a structure’s mass is directly proportional to its volume (i.e., size). Regarding distance, we use the distance to the nearest point on the surface and not the distance to the center of mass, because soft tissue is most influenced by the nearest piece of bone that it is attached to.
>
> Lastly, we would like to emphasize that PolyPose is modular and alternative weight field formulations can be easily integrated. We will emphasize these points in the revised methods section.
>
> > _”With equation (3) and rigidly aligned S_anchor which is available at all views, all angled camera matrices could be estimated, but how many sampled 2D points p in X-ray images are needed for a correct evaluation for the X-ray projection operator P?”_
>
> We sample as many 2D points as there are pixels in the X-ray image. For example, if the X-ray image is $512 \times 512$ pixels, then we cast $512^2 = 262,144$ rays using Equation 3. We note that this rendering operation requires only tens of milliseconds to render a $512 \times 512$ X-ray, as it is efficient and can be parallelized over rays on the GPU.
>
> > _”What is the confidence we should keep for such a discrete approximation of first-order approximation following Beer-Lambert law?”_
>
> A first-order approximation of the Beer-Lambert law is sufficient for registration tasks because
> - The resulting synthetic images are sufficiently similar to real X-rays (see Figure 3); and
> - A discrete renderer implemented in this manner faithfully recapitulates the projective geometry underlying X-ray image formation.
>
> Simulating second-order photonic effects, such as beam hardening or scattering, may further improve small textural details within the rendered X-ray, but these details do not change the image geometry relevant to registration. Lastly, this discrete first-order approximation has been successfully used in many widely adopted 2D/3D X-ray/CT registration approaches (e.g., DiffPose, CVPR 2024 [15]).
>
> > _”Any alternative evaluation loop between camera matrics and acquisition deformation field needed?”_
>
> We are unsure of what the reviewer meant by this question. If they are asking whether it is possible to alternate between optimizing the camera matrices used for polyrigid deformations and the raw unconstrained deformation field itself, this can be done, yes. However, we did not find this to be necessary for our experiments, as the polyrigid deformation widely outperforms baseline methods that operate directly on the deformation field in Table 1 and Figure 5. We would be happy to discuss further in the discussion period if we misunderstood the question.
>
> > _”Equation (9) I can understand the SE(3) to se(3) transformation, but why it is isomorphic to R^6?”_
>
> Thank you for catching this typo. We will update the text to instead say: “the codomain of the logarithmic map is *homeomorphic* to $R^6$” in the revision, where this six-dimensional space corresponds to three rotational and three translational parameters (Appendix B).
>
> > _”I am more interested in a general body data set where different articulated bodies or soft tissues are in different size and mass, and no rigid transform or global alignment guarantee for any body or structure.”_
>
> We respectfully clarify that the Head&Neck dataset exactly satisfies the conditions that the reviewer wished to see tested. Within that dataset, all the articulated bodies and soft tissue structures are of vastly different sizes and masses, including the 21 articulated bodies (L/R humerus, L/R scapula, L/R clavicles, thoracic and cervical vertebrae, and skull) and five soft tissue structures (thyroid, spinal cord, brain, esophagus, and trachea). Furthermore, no global rigid or affine transform describes the motion of patients in this dataset, nor any of their constituent soft tissues.

---

> > ### Comment · Reviewer_Y5jy · 2025-08-07
> > **Confusion resolved.**
> >
> > Thanks for the authors' explanation. All my confusion and concerns regarding the equations and methodology have been addressed. Therefore I decided to keep my positive rating.

---

### Official Review · Reviewer_7gVY · 2025-07-08

**Clarity:** 4
**Significance:** 3
**Originality:** 3
**Rating:** 4
**Confidence:** 5

**Summary:**

This paper introduces a locally-rigid registration framework (PolyPose) for CT to X-ray (2D/3D) registration. PolyPose is a two-stage method. In stage 1, PolyPose estimates the initial camera poses for each X-ray scan with differentiable X-ray rendering and gradient descent. Subsequently, PolyPose leverages segmented rigid structures (bones) to create a precomputed weight field and estimate the local rigid transform for each structure. The final deformation field is a weighted combination of the resulting local rigid transforms. The proposed method was evaluated using synthetic Head&Neck datasets (19 test subjects) and real clinical datasets (DeepFluoro, comprising 6 test subjects with CT-X-ray pairs). PolyPose achieves the best overall registration with no folding in the deformation field among 3D registration methods (uniGradICON, multiGradICON, FireANTs, anatomix) and 2D/3D registration methods (DiffPose, LiftReg and 2D3D-RegNet).

**Questions:**

Can PolyPose register images with serious metal artefacts? Can the differentiable X-ray projection accurately simulate the metal artefact?

Given 1 CT and 1 X-ray scan, will there be any difference between PolyPose and DiffPose?

In what situations will the DiffPose fail but the PolyPose not?

**Ethical Concerns:**

["NO or VERY MINOR ethics concerns only"]

**Final Justification:**

The rebuttal has addressed all my concerns. There is a clear difference between DiffPose and PolyPose. The proposed method is a practical solution for CT to X-Ray registration. The proper combination of differentiable X-ray rendering, rigid structure segmentations and transformation model shows well-grounded effort. Since the proposed method cannot directly generalise to all the 2D-3D registration tasks, its impact is slightly hindered. Hence, I will keep my positive rating ('Borderline accept').

**Limitations:**

PolyPose relies on rigid structure segmentation. The potential limitations and impact are discussed in the paper.

**Paper Formatting Concerns:**

None.

**Quality:**

3

**Strengths And Weaknesses:**

**Strength**

A practical and elegant solution for CT to multi-X-ray registration. This paper proposes a promising solution for CT to multi-X-ray registration, a less-explored yet important topic in the context of image registration. By leveraging strong domain priors (bones are rigid, articulated motion is piecewise rigid) and differentiable X-ray rendering, PolyPose achieves robust and accurate CT to multi-X-ray registration without the need for a large training dataset.

Thorough and rigorous evaluation. The registration performance is comprehensively evaluated with both synthetic and real clinical images. Moreover, an ablation study on the deformation parameterization, weight function and rigid components is provided.

SOTA results in 2D/3D registration. PolyPose achieves superior results over strong competitive methods such as DiffPose and FireANTs.

Strong reproducibility. The source code is provided in the attachment and is well-written.

Good clarity. This paper is well-written. The figures and mathematical formulations are clear and easy to follow.

**Weakness**

Reliance on differentiable X-ray rendering and rigid structure segmentations: PolyPose comprises differentiable X-ray rendering and rigid structure segmentations, which hinders its potential for other applications such as MRI to X-Ray registration. Moreover, the differentiable X-ray rendering is tailored for CT to X-ray projection, and it is not applicable to other modalities, which further limits the application of PolyPose.

Limited novelty. The success of PolyPose has been greatly fueled by differentiable X-ray rendering, rigid structure segmentations, and polyrigid registration, which are not new in the literature. Although an improvement over the weight field is introduced, the performance gain is rather marginal, as shown in Table 3.

Robustness to exceptional cases. PolyPose may fail to handle soft tissue deformation and metal artefacts. First, as discussed in the limitation, since PolyPose assumes rigid bodies capture all motion, the local deformation of soft tissue, such as breathing and muscle contraction, may degrade the quality of the solution. Second, in real clinical scenarios, the metal artefact is common in post-treatment CT scans. Can PolyPose register images with serious metal artifacts? Can the differentiable X-ray projection accurately simulate the metal artefact?

---

> ### Author Rebuttal · Authors · 2025-07-30
>
> We thank the reviewer for their insightful comments and questions. We are glad that they found the problem to be under-explored yet important, and our solution to be practical, elegant, and rigorously evaluated.
>
> > _”PolyPose comprises differentiable X-ray rendering and rigid structure segmentations, which hinders its potential for other applications such as MRI to X-Ray registration..”_
>
> We agree that our work does not apply to the MRI to X-ray registration problem. For context, CTs and X-rays are directly related via the Radon transform, which enables differentiable rendering of 2D X-rays from 3D CTs. In contrast, MRI and X-ray share no analytical relationship, changing the problem to *multi-modal* 2D/3D registration of slices to a volume, which is different from the problem we tackle.
>
> We respectfully posit that, as CT volumes are nearly universally acquired for planning various interventional procedures (e.g., radiotherapy, orthopedic surgery), our problem formulation accepts a broad scope of potential operative applications of PolyPose. We agree that this should be made clearer to the reader and we will revise our discussion to do so. Thank you for the clarifying question!
>
> > _”Limited novelty. The success of PolyPose has been greatly fueled by differentiable X-ray rendering, rigid structure segmentations, and polyrigid registration, which are not new.”_
>
> We agree that the listed sub-components of our framework are foundational concepts in CT and X-ray image analysis. To clarify, we do not claim to contribute new methods for segmentation or X-ray rendering. Our main contribution is a new problem formulation: deformable piecewise-rigid alignment of preoperative volumes to intraoperative X-rays. Quantitatively, our work sets a new empirical state-of-the-art for this under-explored yet important problem (as noted by the reviewer), enabling surgical navigation applications that were not possible with previous methods. We will explicitly clarify this in the revision.
>
> > _”As discussed in the limitation, since PolyPose assumes rigid bodies capture all motion, the local deformation of soft tissue, such as breathing and muscle contraction, may degrade the quality of the solution.”_
>
> This may be a potential miscommunication. While our method does not explicitly represent soft tissue deformations, soft tissue structures directly contribute to the image-based loss used in alignment (Figure 3B) and are thus incorporated into the deformation estimation. To clarify, our only assumption is that bones move rigidly; no rigidity is assumed for other structures.
>
> For example, in the highly ill-posed setting of CT to X-ray registration of the Head&Neck dataset containing both rigid bodies and soft tissue, methods that do not use this prior produce unrealistic and suboptimal results (Figure 4), as soft tissue structures are very difficult to distinguish in X-ray and the deformation model is unconstrained. In contrast, our prior on human motion enables PolyPose to successfully extrapolate the deformations to soft tissue structures that were not explicitly modeled to achieve the best overall alignment (Figure 5).
>
> > _”In real clinical scenarios, the metal artefact is common in post-treatment CT scans. Can PolyPose register images with serious metal artifacts? Can the differentiable X-ray projection accurately simulate the metal artefact?”_
>
> Thank you for raising this important point about real-world deployment. While metal implants cause serious artifacts on CT, the implants are well-defined on X-ray images, and differentiably rendered X-rays accurately match the appearance of real X-rays. For example, in a different context, [16] successfully used such differentiable renderings from CTs with metal implants for rigid 2D/3D registration.
>
> As a result, PolyPose can register images with metal artifacts. However, PolyPose requires segmentations of rigid bodies in the CT and automated segmentation tools like TotalSegmentator [64] may degrade under severe metal artifacts. Fortunately, as shown in our responses to reviewers `Jfu5` and `CMp5`, PolyPose is robust even with degraded segmentations. We will clarify in the revision that experiments on images with metal implants are an important area of future work.
>
> Lastly, if the reviewer is interested in observing an X-ray rendered from a metal-implanted CT, we would be happy to share a code snippet to do so during the discussion period, as figures are not permitted in the rebuttal.
>
> > _”Given 1 CT and 1 X-ray scan, will there be any difference between PolyPose and DiffPose?”_
>
> Yes, DiffPose and PolyPose will produce very different outputs in this scenario:
> - DiffPose will estimate a single rigid transform to align the two images;
> - PolyPose will estimate multiple rigid transforms (one for each rigid structure in the CT) and interpolate them to produce a dense diffeomorphic deformation field.
>
> Broadly, DiffPose and PolyPose do not differ in the number of scans used, but in the number of rigid transforms estimated.
>
> > _”In what situations will the DiffPose fail but the PolyPose not?”_
>
> As DiffPose is a rigid-only 2D/3D registration method, DiffPose will fail whenever there is no *single* global rotation and translation that captures the patient’s motion between their preoperative and intraoperative positions. On the other hand, PolyPose performs *deformable* 2D/3D registration and can accurately capture the complex articulated multi-joint movements common in human movement. Examples include the articulated motion of the skull, spine, and shoulder complex in the Head&Neck dataset (Figure 5) and the independent movement of the femurs relative to the pelvis in the DeepFluoro dataset (Figure 6 and Figure 10 in Appendix E.2).

---

> > ### Comment · Reviewer_7gVY · 2025-08-07
> >
> > Thanks for the detailed response. The rebuttal has addressed all my concerns. There is a clear difference between DiffPose and PolyPose. The proposed method is a practical solution for CT to X-Ray registration. The proper combination of differentiable X-ray rendering, rigid structure segmentations and transformation model shows well-grounded effort for CT to X-Ray registration. Still, since the proposed method cannot directly generalise to all the 2D-3D registration tasks, its impact is slightly hindered. Hence, I will keep my positive rating ('Borderline accept').

---

### Note · Authors · 2025-08-12

We sincerely thank the reviewers for their insightful feedback and time spent reviewing our work. Their constructive suggestions led to interesting discussions and experiments that will be incorporated in the final revision.

We were pleased that reviewers found our submission addresses an important yet under-explored problem with a practical and elegant solution \[`7gVY`\], demonstrating clear novelty \[`Y5jy`, `CMp5`\], and providing state-of-the-art results in complex settings with an anatomically motivated approach \[`7gVY`, `CMp5`, `Jfu5`, `5y16`\] without requiring hyperparameter tuning \[`CMp5`, `Jfu5`\]. They also highlighted the manuscript's clarity and technical rigor \[`7gVY`, `Y5jy`\], as well as its reproducible implementation \[`7gVY`, `CMp5`\].

The reviewers raised many interesting questions to better understand the limitations of our method, including how our model would perform in real-world clinical settings with severe non-rigid deformations \[`7gYV`, `Y5jy`, `5y16`\], and whether our model is robust to input segmentation errors \[`CMp5`, `Jfu5`\].

To address these points, we conducted additional experiments, including:

* **Benchmarking on a dataset with large soft-tissue components:** To further demonstrate generalizability, we conducted new experiments on another dataset containing soft-tissue deformations (*ThoraxCBCT*). In the highly ill-posed sparse-view 2D/3D registration setting, our polyrigid registration framework achieves state-of-the-art registration results even for anatomical structures with completely nonrigid deformations \[`7gYV`, `Y5jy`, `5y16`\]. This new experiment on the free-breathing thorax further supports results on datasets containing diverse soft tissue structures in the head-neck-shoulder complex and pelvis, demonstrating that the polyrigid prior based on skeletal structures enables generalization to non-rigid structures as well.
* **Robustness to segmentation errors:** To investigate robustness to segmentation errors, we conducted additional experiments that simulate segmentation corruption. We found that our model achieves high registration accuracy even under severe segmentation errors that can occur with off-the-shelf segmentation networks like TotalSegmentator \[`CMp5`, `Jfu5`\].
* Reporting runtimes for all methods as suggested by [`CMp5`, `5y16`].

Detailed descriptions of these experiments, as well as point-by-point answers to all other questions, are available in the individual responses below.

---

### Decision · Program_Chairs · 2025-09-17

**Decision:**

Accept (poster)

**Comment:**

Summary: Approach to model deformable CT to X-ray registration by weighted composition of locally rigid bone transforms.  Experiments show SOTA alignment starting with as few as two views.

Strengths: Strong anatomical prior to regularize an ill-posed task.  Gives consistent gains over alternative methods on head-neck and pelvis.  Approach has fast runtime.

Weaknesses: Needs accurate bone segmentations and visible skeletal anchors; therefore not applicable to MRI X-ray alignment. or purely soft-Case of severe metal artifacts/anchor occlusion is partly discussed and not fully validated.

Accept: well-justified method with clear empirical wins in sparse/limited-angle settings and added robustness analyses.

(e) Rebuttal & Discussion: Reviewers asked about soft-tissue generalization, segmentation errors, occlusions/metal artifacts, and runtime; authors added ThoraxCBCT results, a segmentation-corruption study, runtime table, and clarified scope, after which all reviewers maintained positive ratings.  Strong consensus for acceptance from reviewers and myself.